# Ubiquitous short-range order in multi-principal element alloys

Ying Han [1,8], Hangman Chen[2,8], Yongwen Sun[1], Jian Liu[3], Shaolou Wei[4,7], Bijun Xie[2], Zhiyu Zhang [1], Yingxin Zhu[1], Meng Li [5], Judith Yang [5,6], Wen Chen [3], Penghui Cao [2]✉ & Yang Yang [1]✉

Recent research in multi-principal element alloys (MPEAs) has increasingly focused on the role of short-range order (SRO) on material performance. However, the mechanisms of SRO formation and its precise control remain elusive, limiting the progress of SRO engineering. Here, leveraging advanced additive manufacturing techniques that produce samples with a wide range of cooling rates (up to $10^7 \, K \, s^{-1}$) and an enhanced semi-quantitative electron microscopy method, we characterize SRO in three CoCrNi-based face-centered-cubic (FCC) MPEAs. Surprisingly, irrespective of the processing and thermal treatment history, all samples exhibit similar levels of SRO. Atomistic simulations reveal that during solidification, prevalent local chemical order arises in the liquid-solid interface (solidification front) even under the extreme cooling rate of $10^{11} \, K \, s^{-1}$. This phenomenon stems from the swift atomic diffusion in the supercooled liquid, which matches or even surpasses the rate of solidification. Therefore, SRO is an inherent characteristic of most FCC MPEAs, insensitive to variations in cooling rates and even annealing treatments typically available in experiments.

Multi-principal element alloys (MPEAs), usually consisting of three or more components with nearly equal elemental ratios, have received extensive attention due to their excellent mechanical performance and damage tolerance under extreme environments[1–8]. A prominent example is the face-centered cubic (FCC) CoCrNi alloy, which exhibits the highest fracture toughness at the liquid helium temperature ever recorded[9]. MPEAs are considered random solid solutions (RSS) with a high configurational entropy[1,10], thus often referred to as medium/high entropy alloys (MEAs/HEAs), depending on their compositional complexity. However, more recent studies show that the elemental distribution in MPEAs at the atomic scale is not purely random, originating from the increasing significance of the enthalpy term in the Gibbs free energy as temperature decreases[11]. Therefore, the RSS state

is only characteristic of the high-temperature region close to the melting point, at which the entropy term is considered to be predominant. At lower temperatures, local ordering, such as the short-range order (SRO) and local clustering[12–15], will develop in MPEAs to minimize the free energy. Specifically, the concept of tuning SRO in MPEAs has garnered extensive interest[12–14,16–18], not solely for its pivotal function in maintaining the stability of the FCC structure[12], but also for its potential as an innovative parameter to facilitate enhanced properties. For instance, it has been reported that SRO can tune the stacking fault energy in a wide range[12], influence magnetization[19], promote work-hardening[13], enhance resistance to fatigue[20] and irradiation damage[18,21,22], and impact corrosion[23–25]. However, several critical questions must be resolved before harnessing SRO engineering as

[1]Department of Engineering Science and Mechanics and Materials Research Institute, The Pennsylvania State University, University Park, PA, USA. [2]Department of Mechanical and Aerospace Engineering, University of California, Irvine, CA, USA. [3]Department of Mechanical and Industrial Engineering, University of Massachusetts, Amherst, MA, USA. [4]Department of Materials Science and Engineering, Massachusetts Institute of Technology, Cambridge, MA, USA. [5]Department of Petroleum and Chemical Engineering, University of Pittsburgh, Pittsburgh, PA, USA. [6]Center for Functional Nanomaterials, Brookhaven National Laboratory, Upton, NY, USA. [7]Present address: Max-Planck-Institut für Eisenforschung, Düsseldorf, Germany. [8]These authors contributed equally: Ying Han, Hangman Chen. ✉e-mail: caoph@uci.edu; yangyang@alum.mit.edu

a novel facet of defect engineering. First, what is the role of SRO on the properties of MPEAs? Existing literature presents conflicting viewpoints on this matter. A number of investigations have suggested that SRO makes minimal, if any, contribution to the yield strength in FCC MPEAs[26–29], while certain theoretical studies have posited that SRO may significantly bolster yield strength[11,30–32]. Second, is it possible to experimentally adjust the SRO degree in MPEAs across a broad spectrum to customize their properties? It is worth noting that the answer to this latter question is a necessary precondition to resolving the first, as reliable research on the impact of SRO hinges upon our ability to tune SRO in the materials precisely. This paper primarily seeks to address the second question due to its pivotal role, while also shedding light on reconciling the disparate responses to the first question.

To date, the majority of research addressing the second question has been primarily focused on modulating SRO by modifying processing methodologies and/or thermal treatments (Supplementary Fig. 1). Particularly, varying annealing temperatures, from 400˚C to 1200˚C, has been documented to introduce SRO in MPEAs (see Supplementary Table 1 for more details). Conversely, it has been observed that the yield strength of polycrystalline MPEAs adheres to the same Hall-Petch relationship, irrespective of the thermal treatment temperature and duration[26]. And more recently experiments on single-crystal CoCrNi reveal neglectable effects of annealing on yield strength[33]. These seemingly contradictory findings underscore a critical knowledge gap in comprehending the kinetics underlying SRO formation−specifically, does thermal treatment alone serve as an effective strategy for SRO engineering in MPEAs?

Here, integrating an improved transmission electron microscopy (TEM) characterization method, advanced additive manufacturing[34], and atomistic modeling, we reveal the answer to this question to be negative. Our comprehensive investigation shows that SRO forms rapidly during the solidification process, even at high cooling rates, which limits the extent to which subsequent annealing can alter the SRO. This finding has significant implications for resolving the paradox concerning the role of SRO in strengthening FCC MPEAs. The absence of yield strength differences between quenched and annealed materials, as observed in many previous experiments, can result from the similar degree of SRO that predominantly forms in solidification. Consequently, it is crucial to acknowledge that rapidly quenched samples retain an important degree of SRO, thus deviating far from the condition of a random solid solution. Additionally, it is important to note that the impact of SRO on the properties of materials may not conform to a linear or monotonic relationship, thereby implying a complexity that necessitates meticulous quantification of SRO and a wide range of SRO degrees in experimental samples to illuminate a comprehensive overview of the influence of SRO. To overcome the intrinsic constraints of thermal treatment in achieving a wide-ranging adjustment of SRO, researchers should contemplate incorporating alternative strategies in tandem with thermal treatments.

## Results
### Characterization of SRO in FCC MPEAs with various materials processing routes
Given that SRO is more energetically favorable at lower temperatures, it is conceivable that an increase in the cooling rate during the solidification process of MPEAs could help suppress SRO formation and retain the RSS state. To bolster the formation of SRO, intermediate temperature annealing is widely adopted, which is ultimately governed by the competing balance between thermodynamic driving force (which intensifies at lower temperatures) and kinetics (which accelerates at higher temperatures, expediting diffusion and reducing SRO formation time). Presumably, quenched samples are commonly deemed closely akin to an RSS, while annealed specimens are associated with a higher degree of SRO[13,35]. Nonetheless, this notion raises a few pertinent questions. Foremost, while it is plausible that the SRO

degree is diminished in the quenched sample, the precise extent of this reduction remains uncertain. Additionally, the quench rate is processing-history-dependent, with conventional water quenching techniques delivering a cooling rate at the order of around 10 to $10^2\,K\,s^{-1}$, a rate that may still be relatively slow.

To assess how cooling rates and thermal treatments influence SRO in MPEAs, we fabricate MPEAs with a cooling rate spanning seven orders of magnitude using traditional casting and several advanced techniques, including laser-directed energy deposition (LDED) and laser powder bed fusion (LPBF), as illustrated in Fig. 1a. The corresponding processing cooling rates[34,36,37] are around $10^1$ to $10^2\,K\,s^{-1}$, $10^3$ to $10^5\,K\,s^{-1}$ and $10^5$ to $10^7\,K\,s^{-1}$, respectively. Following processing, we also annealed one of the as-cast samples at 900 °C for seven days, a treatment recently reported to induce peak local ordering[28]. Our objective is to contrast the SRO in these materials, thereby discerning the impacts of solidification quench rates and thermal treatment on SRO formation. Moreover, we have selected CoCrNi, CoCrFeNi, and CoCrFeMnNi for comparative analysis, potentially revealing insights into how compositional complexity influences SRO formation. These CoCrNi-based MPEAs were chosen based on their compelling mechanical performance, availability of large experimental datasets, and versatile fabrication techniques[38,39]. TEM, X-ray diffraction (XRD) and atomic-resolution scanning transmission electron microscopy - energy dispersive spectroscopy (STEM-EDX) characterizations have verified that these CoCrNi-based MPEAs maintain a single phase and FCC structure even in the LPBF samples that were formed at the highest cooling rates (Fig. 1 and Supplementary Fig. 2−3). The LPBF and LDED samples generally contain more dislocations than the other samples due to the fast-cooling rates. Also, we observe dislocation cell wall structures that are typical of these additive-manufactured alloys[40]. To confirm chemical homogeneity, we performed STEM-EDX elemental mapping for all samples at two different magnifications (45,000 × and 350,000 ×), offering spatial resolution of 1.1 nm and 0.14 nm, respectively. All samples demonstrated chemical uniformity (Supplementary Fig. 4−12), except for the CoCrFeMnNi fabricated via the LPBF method (Supplementary Fig. 13), which exhibited some local clustering. We hypothesize that these non-equiatomic local segregation areas might exhibit a varying degree of CSRO compared to the equiatomic matrix. However, since our primary objective is to determine the presence (or absence) of CSRO at high cooling rates, rather than comparing the extent of order between equiatomic and non-equiatomic alloys, this variation in SRO does not affect the reliability of subsequent characterizations.

It is crucial to evaluate the degree of SRO in these MPEAs using a technique that can collect bulk-representative data. Previously, methods to characterize SRO include high-resolution STEM (HR-STEM)[14,18], resistometry[33], energy-filtered selected area electron diffraction (EF-SAED)[13,41], atomic-resolution EDX[14,16], X-ray/neutron scattering[42], atom probe tomography (APT)[15,29,43] and more recently atom electron tomography (AET)[44]. A comparison of these techniques is provided in Supplementary Table 2, showing that the EF-SAED method is a suitable candidate based on our selection metrics. In this study, we have enhanced the EF-SAED methodology by integrating sophisticated data-processing techniques and a more robust experimental workflow. This method, named ESQ-SAED (Enhanced Semi-Quantitative SAED), not only obviates the need for a costly energy filter while maintaining a suitable signal-to-noise (SNR) ratio but also provides a promising avenue for more precise analysis. Our method and the resultant characterization results are illustrated in the following.

Thin lamellas for TEM characterization were prepared from the bulk samples, with their bright-field TEM images and corresponding SAED patterns shown in Fig. 1b. All the characterizations were performed with the same instrument and alignment conditions to ensure the robustness of the results. In prior research, diffuse spots at the ½ {3̄11} location of FCC MPEAs electron diffraction on [112] zone axis has

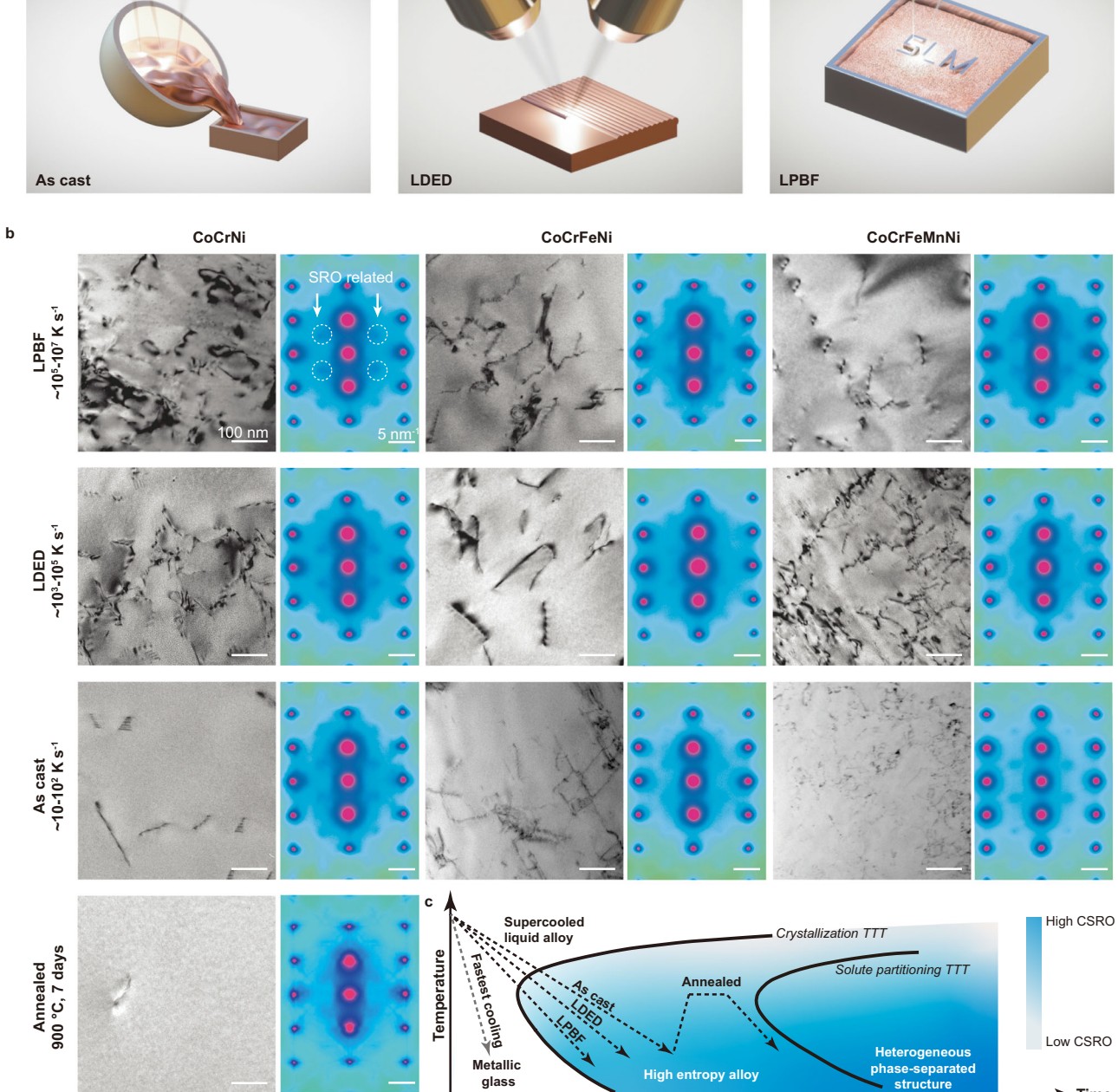

**Fig. 1 | Qualitative characterization of SRO in MPEAs with various thermal processing routes. a** Schematic of the as-cast, LDED, and LPBF sample preparation method. **b** Bright-field TEM images of the selected area and SAED patterns under the same conditions for ten samples. The white arrows and circles indicate the diffused spots related to SRO. **c**, Schematic of the TTT diagram of SRO, showing the difference in SRO is limited even with a wide range of cooling rates. Scale bars for bright-field TEM images, 100 nm. Scale bars for SAED patterns, 5 nm$^{-1}$.

been considered as direct evidence of SRO[13,14] (see more discussions about differentiating the effects of SRO, planar defects and higher-order Laue zones (HOLZ) on the diffuse signals in the section for Fig. 5). Remarkably, the discernible existence of these diffuse spots in the SAED patterns in Fig. 1b reveals that all ten samples exhibit SRO, even the LPBF sample fabricated at the cooling rate of 10$^7$ K s$^{-1}$. A schematic time-temperature-transformation (TTT) diagram is plotted in Fig. 1c to facilitate the visualization of the cooling rate, thermal processing history, and the associated degree of SRO.

To scrutinize the relative degree of SRO among the different samples, we have further improved the above SAED method, which is

schematically illustrated in Fig. 2a–f. 200 SAED images were collected consecutively for each sample with the same imaging condition, such as a fixed exposure time and aperture size. Following this, a composite SAED pattern was constructed by overlaying 10 adjacent SAED patterns to increase the SNR, thereby achieving an aggregated exposure time of 1 second for later analysis. Then, the Bragg spots in each composite SAED image were detected by a cross-correlation algorithm, followed by the classification of Bragg spots into three categories: direct beam, matrix, and SRO. As shown in Fig. 2b, the accurate Bragg spot finding results are visualized by the red disks overlaid on the SAED patterns, sharing the same center as the Bragg

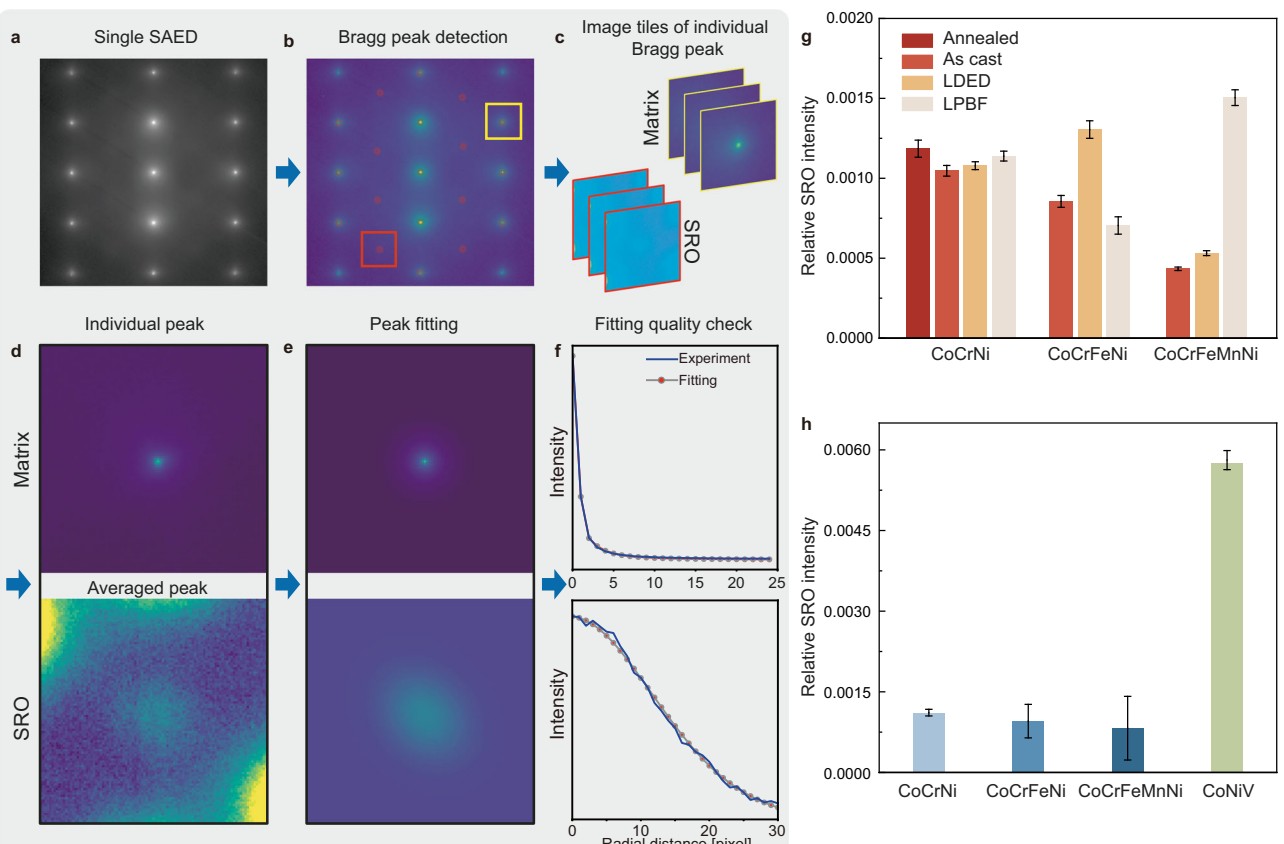

**Fig. 2 | ESQ-SAED characterization of SRO in MPEAs with various thermal processing routes. a–f** Illustration of the ESQ-SAED data processing procedure to enable a more precise analysis of the degree of SRO. Each Bragg peak in the SAED pattern (**a**) will be detected and labeled by the red dots in (**b**). The Braggs peaks will be further classified to create two sets of image tiles, one for the SRO and one for the matrix (**c**). The SNR of the SRO peak is significantly improved by averaging the same type of image tiles, as shown in (**d**). Further, the fitting of the peaks (**e**) and the radial-integral (**f**) facilitate the determination of the peak intensity. The relative SRO intensity is calculated by normalizing the averaged SRO intensity with the averaged matrix intensity. **g** Comparison of the relative SRO intensity of the ten CoCrNi-based MPEAs with different processing methods. **h** Comparison of the relative SRO intensity in four FCC MPEAs with different compositions. The error bar shows the standard deviation.

peaks. Afterward, each SAED image is segmented into a series of square image tiles with the Bragg peak positioned at the center (Fig. 2b, c). Furthermore, the tiles of the same type are averaged to enhance the SNR further, thus alleviating the need for an energy filter. For example, in each SAED pattern, the eight SRO-sensitive tiles closest to the direct beam are selected for averaging. Since there are 20 composite SAED images for each sample, a total of 160 tiles was used to obtain a single image of the SRO Bragg peak (Fig. 2d). The significantly improved SNR is clearly shown by comparing the raw images in Fig. 2c and processed images in Fig. 2d. To further quantitatively analyze the Bragg peak intensities, two-dimensional (2D) Gaussian fitting and pseudo-Voigt fitting are performed for SRO and Matrix peaks, respectively. The corresponding fitting results for both matrix and SRO peaks are shown in Fig. 2e, depicting excellent agreement with the experiments. To validate the quality of the 2D fitting analysis, a radial-integral with elliptical correction is performed on the 2D data (Fig. 2f), which again demonstrates the robustness of the method. Note that the selection of the beam current and exposure time in our experiments was a strategic decision aimed at enhancing the signal-to-noise ratio for the SRO signal and ensuring a close-parallel-beam condition. A more detailed discussion can be found in Method and Supplementary Fig. 14. At last, the SRO Bragg peak intensity divided by the averaged matrix Bragg peak intensity, denoted as the "relative SRO intensity", is computed to compare different MPEA samples with different compositions and thermal treatments (Fig. 2 g, h).

Figure 2g shows the relative SRO intensity of various CoCrNi-based MPEAs with different thermal processing routes. A similar degree of SRO is found in these ten samples, indicating that the variation of cooling rates and heat treatment may not be effective in tuning SRO. To further understand the effect of compositional complexity, we have averaged results for the same composition to obtain composition-specified relative SRO intensities, along with the additional data of a ternary alloy (CoNiV) (Fig. 2h). When the number of elements increases, it is expected that the entropy term in the Gibbs free energy equation ($G = H - TS$, where $G$, $H$, $T$, and $S$ denotes the Gibbs free energy, enthalpy, temperature, and entropy, respectively) will take a more significant role. If the enthalpy term is not significantly changed, then the system would prefer less SRO at the same temperature. Here, our comparison between CoCrNi, CoCrFeNi, and CoCrFeMnNi in Fig. 2h shows that a higher number of constitutional elements seems to be correlated to a reduced intensity of SRO in the SAED patterns. However, the significant error bar indicates that this effect, even exists, may not be substantial. In stark contrast, a considerable increase in SRO is observed in the CoNiV compared to CoCrNi-based MPEAs. This can be rationalized by considering the enthalpy changes led by the choice between V *versus* Cr, Fe, or Mn. Vanadium can form stable or meta-stable intermetallic compounds with Ni and/or Co, thus manifesting a higher tendency to form chemical SRO (CSRO)[14]. Here, our observation of a higher degree of SRO in CoNiV is consistent with previous characterization results by other methods[45], indicating that our semi-quantitative

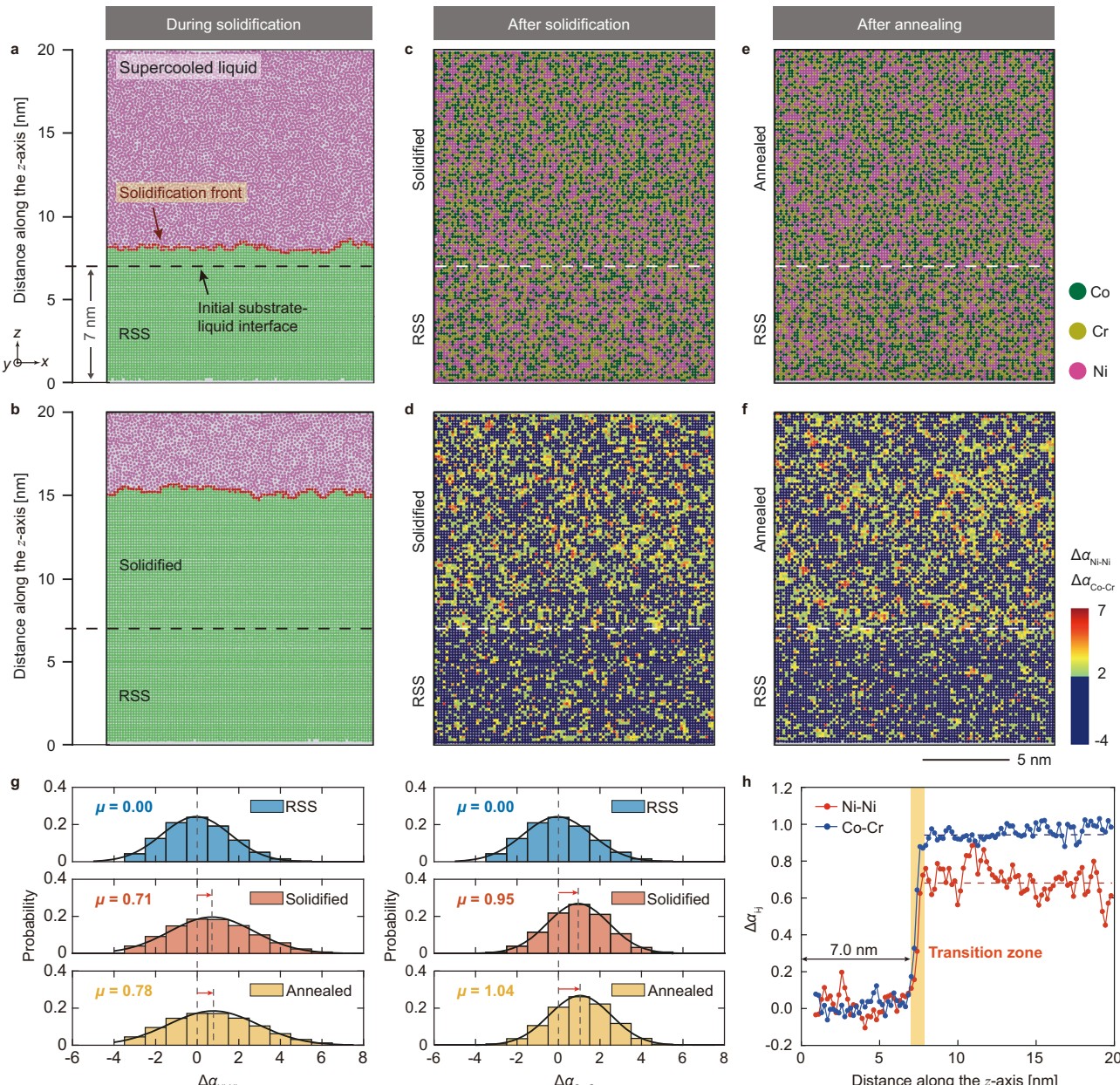

**Fig. 3 | Modeling the evolution of chemical distribution during solidification and after annealing in an equiatomic CoCrNi MEA. a, b** Snapshots showing the solidification process. Green represents the FCC crystal structure, red represents the solidification front, and purple represents the liquid. **c, d** Atomic slices of the structure (in the *x-z* plane) showing the distribution of atoms (**c**) and the pairwise order parameters $\Delta\alpha_{Ni\text{-}Ni}$ and $\Delta\alpha_{Co\text{-}Cr}$ (**d**) after solidification, respectively. A threshold of 2 has been applied on the colorbar of the pairwise order parameters to show the contrast. Thus, the atoms with order parameters smaller than 2 are colored by dark blue. The original result without threshold is included in Supplementary Fig. 16. **e, f** Atomic slices of the structure (in the *x-z* plane) showing the distribution of atoms (**e**) and the pairwise order parameters $\Delta\alpha_{Ni\text{-}Ni}$ and $\Delta\alpha_{Co\text{-}Cr}$ (**f**) after annealing, respectively. **g** The probability distribution of $\Delta\alpha_{Ni\text{-}Ni}$ and $\Delta\alpha_{Co\text{-}Cr}$ order parameters in the RSS, as-solidified and as-annealed samples. **h** Line profiles showing the variation of $\Delta\alpha_{Ni\text{-}Ni}$ and $\Delta\alpha_{Co\text{-}Cr}$ as a function of solid growth distance in the solidifying direction (*z*-axis) in the as-solidified sample.

ESQ-SAED characterization is reliable. Furthermore, our results suggest that choosing the constitutional element (*i.e.*, moderating the enthalpy term in Gibbs free energy) might be more effective in tuning SRO than adjusting the number of constitutional elements.

**Prevalent SRO formation during solidification**
The intriguing SRO characterizations above motivate a bold hypothesis that a large degree of SRO is formed during solidification (crystallization), in which the MPEAs have been presumably considered to be RSS. To test this critical hypothesis and fundamentally reveal the kinetics of SRO formation, we create a molecular dynamics (MD)

model which allows us to reveal the atomistic solidification process and its resultant chemical distribution in an equiatomic CoCrNi MEA (see Methods). This model (Fig. 3a) consists of a random solid solution (RSS) substrate and a supercooled liquid region. The solidification front commences at the substrate-liquid interface (indicated by the dash lines in Fig. 3a–f). During solidification, this front gradually progresses upward, accompanied by growth of solid and depletion of supercooled liquid (Fig. 3a, b). Using this model, we first study the evolution of chemical ordering at an extremely high cooling rate of $10^{10}\,\text{K s}^{-1}$ (Fig. 3c, d), which is 3 to 5 orders of magnitude higher than that in our LPBF samples (Fig. 1). Surprisingly, the spatial distribution

of atoms in the grown solid reveals augmented Ni-Ni and Co-Cr pairings in the solidified region (Fig. 3c, d), indicating a strong chemical non-randomness feature. To facilitate the quantification and visualization of the CSRO, the corresponding spatial distribution of local pairwise order parameters, such as $\Delta\alpha_{Ni-Ni}$ and $\Delta\alpha_{Co-Cr}$ (modified from the Warren Cowley parameters, see Methods, Supplementary Note 1 and Supplementary Fig. 15–16. for more details), is employed. A positive/negative value of these parameters indicates favored/unfavored bonding in the first nearest neighbor shell, while a zero local pairwise order parameter represents random distribution. Our results show that CSRO (revealed by the heterogeneous distribution of local pairwise order parameters in Fig. 3d) can readily form during solidification with an extremely fast cooling rate.

To assess the proximity of the degree of CSRO after solidification to its equilibrium state after long-time annealing, we employed a hybrid Monte Carlo/molecular dynamics (MC/MD) approach to attain the equilibrium state. The results in Fig. 3e, f, show that only a slightly higher degree of CSRO is achieved after annealing, implying the chemical distribution has closely approached the equilibrium even at the fast solidification speed. In Fig. 3g, we compare the probability distributions of $\Delta\alpha_{Ni-Ni}$ and $\Delta\alpha_{Co-Cr}$ in three systems, including the ideal RSS, the rapidly solidified state, and the annealed equilibrium state. With being zero for RSS, the mean values of $\Delta\alpha_{Ni-Ni}$ only increase from 0.71 (as-solidified) to 0.78 (as-annealed) after aging. Similarly, the mean values of $\Delta\alpha_{Co-Cr}$ show a minor increase, from 0.95 in the as-solidified sample to 1.04 in the as-annealed sample. The probability distributions of the other four pairwise order parameters are shown in Supplementary Fig. 17. These results indicate that most of the CSROs are formed during the solidification process and that heat treatment cannot significantly increase the degree of CSRO, even in the samples solidified at a cooling rate of $10^{10}\,K\,s^{-1}$ which was assumed to be RSS by previous studies. It is important to recognize that kinetic frustration[46] could inhibit the enhancement of CSRO in actual materials. Consequently, the level of CSRO observed in our simulations may not represent the absolute maximum that can be achieved under different conditions or in a thermodynamically ideal system.

To reveal spatial variation of CSRO during solidification, we characterize the values of $\Delta\alpha_{Ni-Ni}$ and $\Delta\alpha_{Co-Cr}$ along the solidification direction (as shown in Fig. 3h), mapped as a function of distance from the bottom to top (i.e., the $z$ direction, as defined in Fig. 3a). Both $\Delta\alpha_{Ni-Ni}$ and $\Delta\alpha_{Co-Cr}$ show a rapid increase at the boundary between the RSS substrate and just solidified region (i.e., the initial substrate-liquid interface, as indicated by the dash line in Fig. 3a), above which the chemical order quickly reaches a quasi-steady value within six atomic layers. This narrow transition zone suggests that the CSRO mainly originates from the crystallization processes, and the substrate has a negligible effect on the degree of its formation. To investigate the influence of the cooling rate on the CSRO formation, we further simulate the solidification at an even higher cooling rate of $10^{11}\,K\,s^{-1}$. Although the cooling rate is 10 times higher, the formation of CSRO is still inevitable during the solidification of CoCrNi MEA. More details can be found in Supplementary Note 2, Supplementary Table 3, and Supplementary Fig. 18, 19. In addition, we repeated the solidification modeling using a different force field, which resulted in a similarly large degree of CSRO during solidification (see Supplementary Note 4).

## Formation kinetics of CSRO at the solidification front

CSRO formation during solidification is a compelling process that warrants further in-depth study. In the supercooled liquid (amorphous structure), we have found that the same CSRO does not appear, while the structurally ordered local clusters, such as the icosahedron, exist (Supplementary Note 3 and Supplementary Fig. 20–22). Considering that crystal growth relies on the advancement of the solidification front (Fig. 3a), it is reasonable to hypothesize that CSRO

predominantly emerges at the phase transformation interface under non-equilibrium solidification conditions. We carefully examined the evolution of the solidification front by analyzing snapshots taken at 4 ps intervals (Fig. 4a, b). Specifically, Fig. 4a, b reveal the spatial and temporal evolution of structure and chemistry at the solidification front, respectively. The solidification front is rugged and exhibits heterogeneous movement along the growth direction (i.e., the $z$-axis). As time elapses, the segments on the interface move upward, indicating a liquid-solid phase transition. Intriguingly, some segments can retract by moving downward, indicating the existence of local remelting. This could occur as the newly solidified clusters, being thermodynamically unstable due to the presence of dangling bonds, reconfigure locally in search of more energetically favorable pairs, thus lowering the potential energy, as revealed by the black circles in Fig. 4b. To quantify the dynamic motion of the front, we measured the net movement distance per 4 ps along the interface (Fig. 4c). A positive value represents a net growth of solidification, while the opposite indicates remelting. Figure 4c shows clearly that the solidification front moves in a heterogeneous manner. The segments of the solid-liquid interface that undergo crystal growth at the current moment may partially remelt in subsequent moments and vice versa. Yet, it is surprising that the detailed balance of local growth and remelting manages to constrain the roughness of the solidification front to just a few atomic layers, ensuring minimal variation in the interface height to reduce its surface area.

The fast cooling rate and rapid solidification speed pose a question about the kinetic processes that can facilitate such rapid CSRO formation. The decisive question is: "Is the diffusion of atoms in surrounding liquid fast enough to endow the chemical ordering during solidification?" To answer this, we compute the crystal growth (solidification front) velocity and compare it with the atomic diffusivities in liquid. The crystal growth velocity is determined by the average growth distance of the solidification front as a function of time (Fig. 4d). The average growth velocity is $9.45\,nm\,ns^{-1}$, or $9.45\,m\,s^{-1}$, which is significantly faster than the upper limit of crystal growth velocity that can be achieved in normal experiments. The diffusivities of Ni, Co, and Cr in liquid are estimated at $2.06\,nm^2\,ns^{-1}$, $1.86\,nm^2\,ns^{-1}$, and $1.65\,nm^2\,ns^{-1}$, respectively, derived from the curves of mean square displacement as a function of time (Fig. 4e). In light of crystal growth velocity (one-dimensional) and diffusivity (three-dimensional), we illustrate and compare the atomic diffusion range and the average distance traveled by the solidification front in the $z$-direction (i.e., the crystal growth distance) over 37 ps (Fig. 4f). When the crystal grows for 0.35 nm (~2 atomic layers), the corresponding diffusion distances of Ni, Co, and Cr are 0.28 nm, 0.26 nm, and 0.25 nm, respectively. The similarity in the crystal growth distance and the atomic diffusion domain sizes over the same amount of time indicates that the diffusivities in the liquid are high enough to provide the source of atoms for the formation of chemical ordering locally. An additional example using a higher cooling rate is provided in Supplementary Note 2 and Supplementary Fig. 19.

We propose an atomistic mechanism for the CSRO formation during solidification, as schematically shown in Fig. 4g–j. The randomly distributed atoms in the liquid adjacent to the solid front diffuse and participate in the growth of the solid. Due to the attractive interactions between atoms in the crystal solid, the favored pairs form and lead to chemical ordering, for instance, a precursor of Ni–Ni pair (Fig. 4g). When unfavorable atoms enter the first nearest neighbor shell of this Ni precursor (for instance, Co and Cr, as shown by the arrows in Fig. 4h), they can depart from the precursor either by interface diffusion or dissolution into liquid (i.e., remelting), leaving room for extra Ni atoms to diffuse in and grow the preferable Ni–Ni pairs. The subsequent growth of Ni-Ni pairs depicts the formation of local chemical ordering of Ni–Ni (Fig. 4i). Once the Ni atoms within the accessible diffusion range are exhausted, the Co and Cr atoms

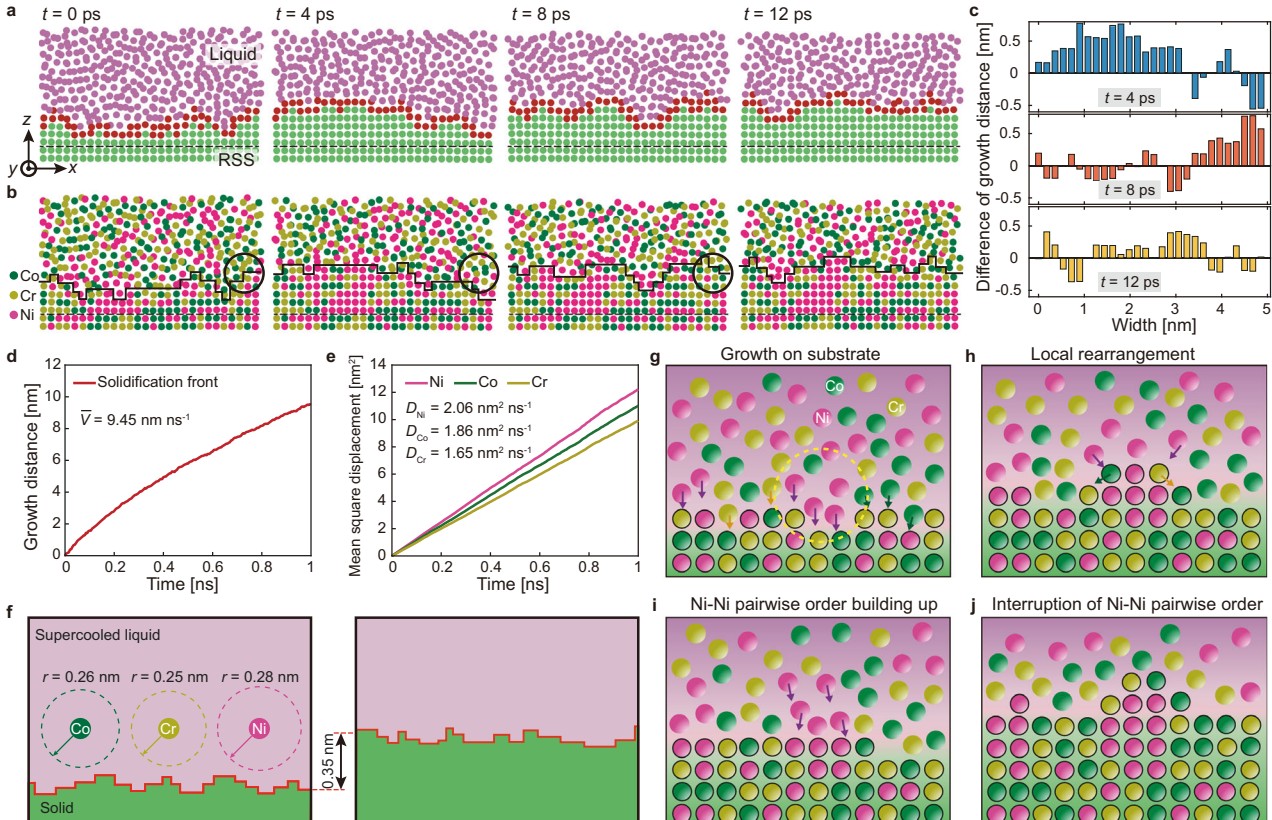

**Fig. 4 | Atomistic insights into the formation mechanism of CSRO at the solidification front. a, b** Snapshots of the solidification front evolution. The top and bottom rows highlight the structural and chemical evolutions, respectively. **c** The net movement distance (in the growth direction, i.e., the $z$-direction) of the solidification front for $t = 4$ ps, 8 ps and 12 ps. The horizontal axis (Width) is equivalent to the $x$-axis, as defined in the coordinate system in (**a**). **d** The solidification front's growth distance as a function of time. The average growth velocity is 9.45 nm ns$^{-1}$. **e** Mean square displacements of elements in the supercooled liquid as a function of time. The diffusivities for the three elements are labeled. **f** Illustration of the diffusion distance of elements in the liquid (left) and the average distance traveled by the solidification front (right) after the same duration of 37 ps. **g–j**, Schematic illustration of the CSRO formation mechanism and the suppression of long-range chemical order during solidification.

eventually encapsulate the Ni−Ni nanocluster; thus, the chemical order cannot extend over a long range (Fig. 4j). The repeated local diffusion, solidification, and remelting (Fig. 4g–j) act as the engine to sustain the local atomic reconfiguration at the solidification front, leading to the formation of CSRO even at a high cooling rate. While our atomistic simulations focus on the formation of CSRO within the FCC CoCrNi system, the findings bear broader implications when combined with our experimental results, extending their relevance to other FCC MPEAs. The high concentrations of principal elements in MPEAs provide the abundant sources required for CSRO, and the swift atomic diffusion facilitates the process. Originating from the attractive/repulsive interactions among the constituent elements, the chemical order and its large extent are primarily determined by the crystal growth velocity and the diffusivities of atoms. A discussion about the thermodynamic origin of local ordering is provided in Supplementary Note 4, Supplementary Fig. 23−25, and Supplementary Table 4−6.

Recent research has shown that the type of element introduced into the system plays a more critical role than the number of constitutional elements in modulating crystal growth velocity[27]. For instance, adding Cr can increase the growth activation energy, reducing the crystal growth velocity. The solid/liquid interfacial structure also impacts crystal growth velocity, particularly for alloys exhibiting a B2 structure. A higher degree of pre-ordering in the liquid phase ahead of the solidification front can increase crystal growth velocity[28]. Conversely, the diffusivities of individual elements, influenced by component interactions, can be decreased by adding Cr, which raises the diffusion activation barrier. Hence, our study underscores the importance of considering the type of elements incorporated into MPEAs and their interactions when analyzing and predicting CSRO behavior and its subsequent impact on material properties.

## Mechanical tuning of SRO

To investigate additional methods that could be integrated with thermal treatment and quench rate selection for enhanced SRO tuning, we have conducted in-situ nano-mechanical experiments using the ESQ-SAED method. This allows us to elucidate the impact of mechanical deformation and dislocation slip on SRO. CoNiV was selected for this demonstration as it offers a better SNR in its SRO signal (Fig. 2h), which enables more precise quantification of the mechanical deformation effect. We fabricated a push-to-pull (PTP) setup[47] from bulk CoNiV using a focused ion beam (FIB), as depicted in Fig. 5a, b. This device offers a straightforward implementation of tensile testing on a thin sample within the limited space of a TEM column while ensuring stability throughout the loading process. Figure 5c illustrates that the ESQ-SAED characterization was performed on the region circled ahead of the prefabricated crack tip at which the strain concentrates. The loading curve is presented in Fig. 5d. It is important to note that the entire electron transparent region maintained a single-crystal structure without visible planar defects during the experiments. Previous studies have noted that SRO can be destroyed by dislocations gliding across it[11,17,33,48]. In the case of cyclic loading, dislocations are repetitively emitted from the crack tip and migrate away from it during loading, while some of the dislocations are fully or partially recovered in the subsequent unloading. The net positive dislocation emission and

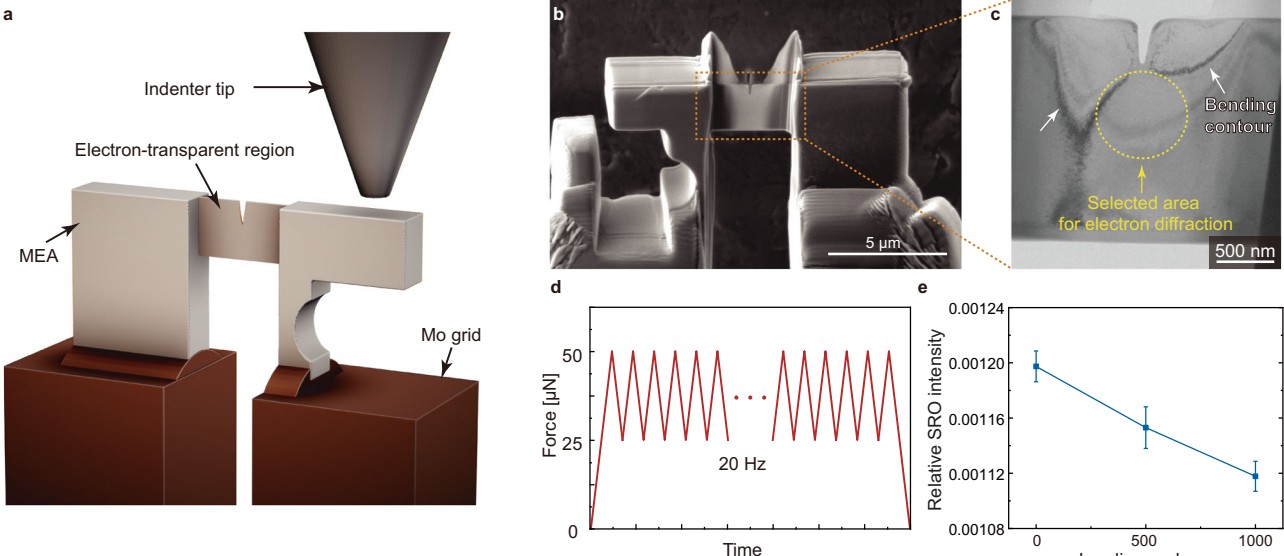

**Fig. 5 | In-situ nano-mechanical testing to understand the evolution of SRO during cyclic loading. a** Schematic drawing of the push-to-pull (PTP) device used for cyclic loading. **b** SEM image of the PTP setup. **c** TEM image showing the boxed region in (b). **d** The mechanical loading curve. **e** Evolution of the relative SRO intensity as a function of loading cycles, underscoring the impact of dislocations on SRO during cyclic loading. The error bar shows the standard deviation.

gliding in each cycle allow us to study its effect on SRO. Aligning with this theoretical speculation, we observed a consistent decrease in the relative SRO intensity as the number of cycles increased (Fig. 5e). A comparison of the relative SRO intensity in all samples used in this study is shown in Supplementary Fig. 26. Compared to thermal treatment and changing the cooling rate, mechanical tuning of SRO might be more controllable.

Moreover, our experiment confirms the ability of the ESQ-SAED method to capture SRO information in MPEAs semi-quantitatively, further reinforcing its effectiveness and reliability. Recent studies have proposed that the diffuse signals at the ½ {3̄11} location of FCC MPEAs electron diffraction on [112] zone axis (a key aspect of our ESQ-SAED approach) have alternative origins such as stacking faults[49] or HOLZ[50]. In the case of stacking faults, we would expect an increase in diffuse signals during cyclic loading, due to the accumulation of planar defects introduced by plastic deformation. Conversely, if these signals were primarily a result of HOLZ, no significant changes should be observed during the cyclic loading of MEAs. Only the decrease in the SRO due to mechanical deformation aligns with our findings that the relative intensity of the diffuse signals reduces during cyclic loading. Therefore, our in-situ TEM experiments provide evidence of a correlation between the diffuse spots on the [112] zone axis and the degree of SRO in the materials.

## Discussion

Prior research has provided clues regarding the prevalence of SRO in FCC MPEAs. For example, Li et al. employed resistometry to examine the evolution of SRO during mechanical deformation in single crystal CoCrNi samples[33]. Two sample conditions were examined: a water-quenched sample from 1473 K (representing minimal ordering) and a sample subsequently annealed at 673 K for 504 h (representing maximal ordering). Interestingly, both samples exhibited a reduction in resistance within the 0–6% engineering strain range, with the decrease in resistance during straining attributed to the destruction of SRO by dislocation movements. Remarkably, both samples displayed a similar maximum decrease in resistivity of 3–4% at 6% engineering strain. An important, albeit unmentioned, inference from these results is that the degree of SRO in the quenched sample might be comparable to that of the annealed sample. For another example, thermodynamics analysis

of binary solid solutions has pointed out that the degree of SRO is a finite (non-zero) value in the high-temperature region[51].

In this study, we demonstrate both qualitatively and semi-quantitatively, through comprehensive electron microscopy characterizations, the ubiquitous nature of SRO in CoCrNi-based MPEAs, regardless of the cooling rates (ranging from $10^1$ to $10^7$ K s$^{-1}$) and even after prolonged annealing periods. In addition, our atomistic simulations of CoCrNi solidification reveal that a substantial degree of SRO can form even under an extreme quench rate of $10^{11}$ K s$^{-1}$ at a temperature close to melting. This rate is a minimum of three orders of magnitude higher than the maximum cooling rate achieved in our experimental setup. Our simulations further elucidate that the extent of SRO is primarily determined by the interplay between the crystal growth velocity and the atomic diffusivities within the supercooled liquid. These factors can be intricately fine-tuned by adjusting the cooling rate and solidification temperature. We found that as the cooling rate increases, the degree of SRO decreases correspondingly, due to the increased solidification front velocity and reduced diffusion range. Considering that the cooling rates employed in experiments to fabricate bulk samples typically do not exceed $10^7$ K s$^{-1}$, our results suggest that a higher degree of SRO, approaching the ideal degree achievable through long-time annealing, is already established after solidification. This implies a small difference of the degree of CSRO between samples subjected to varying thermal treatments, which aligns well with our experimental observations. Therefore, our experimental results, together with modeling outcomes, offer evidence of the ubiquitous nature of SRO in FCC MPEAs.

The inescapable formation of SRO in FCC CoCrNi-based MPEAs carries significant implications for resolving the conundrum surrounding the impact of SRO on yield strength in these alloys. In the realm of body-centered cubic (BCC) MPEAs, both experimental[17] and theoretical[52–56] evidence demonstrate that SRO can modify their strength. However, the scenario with FCC MPEAs is somewhat different. Theoretical predictions suggest a similar effect on strength due to SRO[11,30–32], yet several experimental observations do not corroborate these expectations[26–29]. Specifically, no discernible difference has been observed between the yield strengths of samples with SRO and those considered RSS samples in most experiments. Hence, the manifestation of SRO's role in FCC MPEAs' yield strength remains a challenging

puzzle to untangle. Our research indicates that the previously utilized RSS control sample may, in fact, contain a significant degree of SRO. This presence of SRO calls into question the validity of the comparisons made in past studies. To truly elucidate the impact of SRO on the yield strength of the alloy, it is essential to first procure a sample that accurately represents an RSS state, one that is more devoid of SRO. This approach will establish a more accurate baseline for subsequent investigations.

However, acquiring an actual RSS presents a formidable challenge, mainly due to our limited understanding of effective methods for tuning SRO. Our experiments and modeling results in this study suggest that tuning the cooling rates during solidification and the heat treatments, contrary to previous assumptions, do not efficiently modify SRO, at least for the equimolar CoCrNi-based FCC MPEAs with concentrated solid solutions. Furthermore, the absence of an experimentally verified TTT diagram for describing the kinetics of SRO formation makes its modulation via thermal treatment challenging.

In fact, the relationship between SRO and material properties can exhibit non-monotonic and complex patterns. For instance, the negligible difference in yield strength between quenched and annealed MPEAs could imply a plateau stage. However, there might be a preceding stage characterized by a rapid increase in yield strength, which remains undiscovered due to the lack of an RSS MPEA sample experimentally. The ability to adjust SRO across a broad spectrum is crucial to fully probe this relationship. If the tuning range of SRO is limited, we might only explore a local or specific behavior of the material properties, potentially missing critical turning points that only exist at specific levels of SRO. Hence, the broad tunability of SRO not only allows for a more comprehensive understanding of the SRO-property relationship but also facilitates the identification of optimal SRO levels to achieve desired material properties.

Navigating these complexities and addressing the above challenges may necessitate exploring alternative strategies to modulate SRO, such as mechanical deformation, element substitution, irradiation, and interstitial alloying. As illustrated in Fig. 5, mechanical modulation[33,41] of SRO based on their interaction with dislocations, stacking faults, and twins might be considered. Also, element substitution, for instance, has proven effective in enhancing the formation local ordering. A prime example is the replacement of Mn with Pd in CoCrFeMnNi, which, as demonstrated by atomic Energy Dispersive X-ray Spectroscopy (EDS) mapping, has significantly promoted local ordering[16]. In addition, irradiation has demonstrated its potential to both augment and diminish SRO[57,58]. This is due to the complex interaction between radiation-enhanced diffusion and atomic reshuffling during the irradiation-induced thermal spike. Fine-tuning irradiation parameters could thus provide an innovative approach to effectively manipulate SRO in MPEAs. Moreover, recent discoveries suggest that introducing spatially dispersed interstitial elements such as carbon (C) and nitrogen (N) can facilitate the formation of SRO[59]. This can be attributed to their varying chemical affinities with the principal elements. Integrating one or more of these alternate approaches with a careful control of the cooling rate and heat treatment could pave the way for more precise tuning of SRO in a wide range. This, in turn, could lead to more reliable experiments to understand SRO's influence on material properties and performance.

Additionally, understanding the impact of SRO on material properties necessitates a precise quantification of SRO due to the potentially non-monotonic nature of this relationship. A non-monotonic relationship suggests that small variations in SRO may significantly alter material properties in complex and potentially unpredictable ways. Therefore, without accurate measurement of SRO, it would be challenging to map this intricate relationship accurately, leading to misconceptions about the role of SRO. The ESQ-SAED

method we developed in this study is more quantitative than the previous EF-SAED method, and it negates the need for an energy filter, which considerably simplifies its application. However, this method remains semi-quantitative due to the inherent challenges in calibrating the exact degree of SRO. The precise quantification of SRO in MPEAs persists as a key hurdle in understanding the influence of SRO on material properties. It is crucial for future investigations to pioneer a more accurate quantification method, which could greatly expedite the exploration and application of diverse SRO modulation techniques. Such an advancement, though challenging, might be achievable through the integration of theoretical models grounded in a comprehensive understanding of interatomic interactions. In addition, the distinction between the increase in SRO and the concurrent rise in other defect types and configurations deserves further in-depth study. A discussion on this topic is provided in Supplementary Note 5.

In conclusion, the ubiquitous presence of SRO in FCC MPEAs, attributable to its efficient formation during the solidification process across a wide cooling rate range, challenges the efficacy of thermal treatment as a method to modulate SRO. Consequently, future research should invest more in a combination of multiple potential factors influencing SRO formation, as well as understanding the nuanced impacts of SRO on specific properties of MPEAs. This continued research is vital to unlocking new pathways to manipulate material properties and expanding our comprehension of these complex alloys.

## Methods
### Bulk sample preparation
Samples, including CoCrNi, CoCrFeNi, and CoCrFeMnNi, were prepared by traditional casting, LDED, and LPBF, respectively. The as-cast samples with the same composition were prepared by arc-melting constituent elements with 99.99% purity under an argon atmosphere. The cast ingot was flipped and remelted multiple times to ensure chemical homogeneity. The annealed CoCrNi sample was sealed in a quartz tube with argon and then heat-treated at 900 °C for 7 days. LDED was conducted using an Optomec Laser Engineered Net Shaping (LENS) 450 system with an IPG fiber laser (400 W) at a wavelength of 1064 nm and a laser spot size of ~400 μm at the focal point. During the printing process, the chamber was filled with high-purity argon as an inert gas to keep the oxygen level below 20 ppm. The LDED typically relies on feeding powders into the melt path and molten pool created by a laser beam to deposit material layer-by-layer upon a substrate part or build plate. The LPBF samples were prepared using a commercial M290 (EOS) LPBF machine with a maximum power of 400 W and a focal diameter of 100 μm. All samples were prepared in an argon environment with an oxygen concentration of less than 1000 ppm. The part is formed by spreading thin layers of these powders and fusing layers upon layer of these powders under computer control. The typical cooling rates for LPBF, LDED, and traditional casting are about $10^5$ to $10^7 \, K \, s^{-1}$, $10^3$ to $10^5 \, K \, s^{-1}$, and $10^1$ to $10^2 \, K \, s^{-1}$, respectively. The CoCrNi and CoCrFeNi samples labeled as LDED were laser-scanned as-cast samples using a LENS system. The LPBF CoCrNi samples were laser-scanned as-cast samples using an EOS M290 system. Only the top layers of all samples were used for analysis. The process parameters of LPBF and LDED are summarized in Supplementary Table 7. XRD characterization of the bulk samples was performed on a Malvern Panalytical Empyrean III diffractometer. The bulk VCoNi samples for the in-situ TEM experiments were produced via arc melting of the constituent elements. This process was succeeded by cold rolling, reducing the thickness by 55%. The samples then underwent a homogenization process at 1200 °C for 5 h within an argon atmosphere before being rapidly cooled in ice water. A subsequent round of cold rolling was carried out, leading to a further 65% reduction in thickness. Finally, the sample was annealed at 1000 °C for 10 min and then rapidly quenched in ice water.

## TEM sample preparation

The samples underwent a TEM sample preparation procedure by lifting them out from the bulk sample and welding them on gold half-grids using the Thermo Fisher Scientific (TFS) Helios Nanofab 660 FIB. It is important to note that, for the LDED and LPBF samples, the lifting out of the sample was performed on the newest printed layer. This choice ensured that the selected region had not undergone any additional annealing processes. After the lift-out, the samples were thinned by Ga$^+$ FIB working at 30 kV, which was subsequently reduced to 16 kV and then 8 kV. This process reduced the sample thickness to ~240 nm. The samples used for Figs. 1–2 was then flash polished[60,61] to about 100 nm to remove the Ga$^+$ damaged layer, i.e., about 70 nm of thickness was removed from each side. Based on our Monte Carlo simulation of Ga$^+$ implantation in CoCrNi, CoCrFeNi, and CoCrFeMnNi (Supplementary Fig. 27), the removed layer (~70 nm on each side) is much thicker than that of the damage layer induced by Ga$^+$ ions during the FIB process (typically below 5 nm). The setup for flash polishing is shown in Supplementary Fig. 28a, b. The samples (Supplementary Fig. 28c, d), held by gold tweezer, were immersed in 4% perchloric acid/ethanol solution at 12 V and around −45 °C. The polishing time was controlled by a timer and typically lasted between 50 ms and 200 ms, depending on the sample thickness. After flash polishing, the samples were washed three times in −45 °C methanol, −45 °C ethanol, and 25 °C ethanol, respectively. The resulting sample after flash polishing is shown in Supplementary Fig. 28e, f. Note that the samples used for in-situ mechanical testing in Fig. 5 have not been flash-polished because it might compromise the stability of the mechanical testing process by etching the PTP structure or altering the crack shape, which is undesirable. The crack was deliberately created to localize the deformation zone for a more precise capture of the SRO-deformation relationship. A discussion of the FIB damage in this sample used in Fig. 5 is provided in Supplementary Note 6.

## TEM characterization and analysis

The SAED patterns were collected in TFS Talos F200X TEM at the same alignment and acquisition conditions on the same day, including an accelerating voltage of 200 kV, a spot size of 1, a screen current of 0.03 nA (22,500 × magnification, with the 40 μm select area aperture inserted), a select area aperture of 40 μm, a magnification of 22,500 ×, a camera length of 410 mm and an image resolution of 1024 × 1024. The exposure times for the data in Figs. 1, 2, and 5 are 1 s, 0.1 s, and 1 s, respectively. We have applied a scaling factor for obtaining the relative SRO intensity in Fig. 5, as explained in Supplementary Note 7 and Supplementary Fig. 29. The relationship between beam current and C2 strength for our TEM is measured and shown in Supplementary Fig. 14. The C2 strength was selected to approximate parallel beam conditions, ensuring sharper and more reliable SRO peaks. The details of the ESQ-SAED method have been presented in the main text. The data processing was performed with the py4DSTEM package[62]. The fitting ranges for SRO and Matrix peaks are $r \in [0, 30]$ pixels and $r \in [0, 25]$ pixels, respectively. The anisotropic strain within our samples may contribute to either isotropic or anisotropic broadening of the Bragg peaks in SAED patterns; however, these factors do not detract from the overarching conclusions of our research, as discussed in Supplementary Note 8 and Supplementary Figs. 29–32. The in-situ TEM mechanical experiments were performed with a Bruker pico-indenter (PI) 95 holder.

## Monte Carlo simulation of ion irradiation damage caused by FIB

The simulation is performed by a software package called Ion Irradiation in Materials 3D (IM3D)[63], which is a Monte Carlo simulation code based on the binary collision approximation. We used the full-cascade mode to predict the vacancy distribution of primary radiation damage. By approximating the ion milling angle in a typical FIB thinning process (i.e., a glancing angle of 2°, as shown in Supplementary Fig. 27a), the simulation of Ga$^+$ ion damage is performed for three different CoCrNi-based alloy systems at 5 different Ga$^+$ ion energies (Supplementary Fig. 27b–d).

## Solidification modeling

To investigate the formation of CSRO in CoCrNi MEA, we established bulk models with [100], [010], and [001] orientations along the $x$, $y$, and $z$ directions, respectively. The simulation cells comprised 50 × 50 × 60 unit cells and contained 600,000 atoms. The three elements had equal atomic ratios and were randomly distributed. A many-body embedded atom method (EAM) potential[11] was employed in the simulations of CoCrNi MEA for Figs. 3–4. Also, we used a different EAM potential[4] to validate the conclusion further, as discussed in Supplementary Note 3–4. Periodic boundary conditions were applied along the $x$ and $y$ dimensions of the model, and the non-periodic boundary condition was used along the $z$ dimension. The models were thermally equilibrated at 1315 K and 1015 K by an anisotropic zero-pressure isobaric-isothermal NPT ensemble for 10 ps with an MD time step of 0.001 ps. Next, we fixed the atoms in the bottom 1/3 along the $z$-axis, and the rest was heated up to 2000 K for 50 ps for complete melting. The solid/liquid dual phase systems were cooled down from 2000 K to 1415 K (the melting point temperature) within 50 ps. The pinned atoms were released, and the system was relaxed at 1315 K and 1015 K, respectively, by the microcanonical ensemble (NVE). The Nose-Hoover temperature-rescaling thermostat was applied to the initially fixed solid phase to absorb heat from the supercooled liquid and cool the system to 1315 K and 1015 K, respectively. The cooling rates were estimated by the difference in initial and final temperature of the initially liquid region over solidification time. The Large-scale Atomic/Molecular Massively Parallel Simulator (LAMMPS) was utilized to carry out MD simulation[64], and atomic structures were visualized by the molecular visualization package OVITO[65].

## Local chemical short-range order parameter

The local short-range order was calculated as the nonproportional number of local atomic pairs $\Delta\alpha_{i-j}$, modified from the Warren-Cowley parameter[12,66,67]:

$$\Delta\alpha_{i-j} = N_{i-j} - N_{i-j,0} \tag{1}$$

where $N_{i-j}$ is the number of $i$-$j$ pairs in the first nearest neighbor shell of an $i$-type atom, $N_{i-j,0}$ is the number of $i$-$j$ pairs for the random solid solution. Since there are 12 first nearest neighboring atoms for FCC structure, an equimolar CoCrNi would give rise to $N_{i-j,0} = 4$ (i.e., 12 × 1/3 = 4). For species $i$ and $j$, a positive $\Delta\alpha_{i-j}$ suggests the clustering in the first shell, and the negative $\Delta\alpha_{i-j}$ means the opposite. The difference between this definition of SRO parameters and the more commonly used Warren-Cowley parameters is provided in Supplementary Note 1, showing that the definition of SRO parameters will not change the conclusion and reliability of this work.

## Solidification front analysis and crystal growth

The crystal growth velocity is estimated by tracking the average locations of atoms in the solidification front as a function of time. Both the RSS substrate and the solidified region have FCC structures. Supercooled liquid region has amorphous structures. The solidification front is the interface between the solidified crystal and the surrounding liquid. The solidification front is identified by locating atoms that belong to the FCC crystal structure and are also connected to the liquid phase.

## Ion diffusivity in liquid

The diffusion of elements in the liquid is characterized by diffusion coefficient $D$ via the equation,

$$D = \frac{\langle \mathbf{R}^2 \rangle}{2dt} \tag{2}$$

where $d$ is the dimension of diffusion and is equal to 3 in our case and $t$ is the time. $\langle \mathbf{R}^2 \rangle$ is the mean square displacement (MSD) of an ensemble of particles over time that can be written as follows:

$$\langle \mathbf{R}^2 \rangle = \frac{1}{N} \sum_{i=1}^{N} |\boldsymbol{r}_i(t) - \boldsymbol{r}_i(0)|^2 \tag{3}$$

where $N$ is the number of element in the system, $\boldsymbol{r}_i(t)$ is the position of atom $i$ at time $t$ and $\boldsymbol{r}_i(0)$ is the initial position of atom $i$. We performed independent diffusion simulations to ensure that the measurement of diffusivity is not influenced by solidification. We created a system with a dimension of $20a_0 \times 20a_0 \times 20a_0$ ($a_0$ is lattice constant) that contained 32, 000 atoms. Periodic boundary conditions were applied to three dimensions. The system's temperature was increased to 2000 K for 10 ps to melt the crystal completely and then relaxed at target temperature for 10 ps. Afterward, the MSD of each element was calculated by Eq. 3 every 1 ps. The square root of diffusivity was computed to reflect the particle diffusion distance over a time:

$$\boldsymbol{r} = \sqrt{Dt} \tag{4}$$

where $D$ is the diffusivity and $t$ is the time.

### Degree of SRO at thermodynamic equilibrium

The ideal CSRO through infinite time annealing could be generated by hybrid MC/MD simulation using the VCSGC package[68] under the semigrand canonical ensemble at 1315 K and 1050 K, respectively. 100 MD steps were carried out between MC cycles, with $0.2 \times N$ trial moves for each MC cycle, where $N$ is total number of atoms. The chemical potentials $\Delta\mu_{\text{Ni-Co}} = 0.021$ eV and $\Delta\mu_{\text{Ni-Cr}} = -0.31$ eV derived from the semigrand canonical ensemble simulation were used. The variance parameter $\kappa$ used in simulations was 103. Periodic boundary conditions were applied to all directions of the system. The MD timestep was set to 0.0025 ps, and 1 million MD steps were carried out to achieve converged SRO. It is worth noting that the MC method, which is unable to model the kinetics of SRO formation, aims to create a thermodynamic equilibrium state. To uncover the atomistic processes leading to ordered clusters, one should adopt the kinetic MC method[69].

## Data availability

The main data supporting the findings of this study are available within this article and its Supplementary Information. Additional data are available from the corresponding authors upon request.

## Code availability

py4DSTEM is an open-source package available on GitHub: https://github.com/py4dstem/py4DSTEM.

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

## Acknowledgements

Y.Y. and Y.H. acknowledge support from the National Science Foundation early career award DMR-2145455. The simulation and modeling work done by P. C., H.C., and B. X. was supported by the U.S. Department of Energy (DOE), Office of Basic Energy Sciences, under Award No. DE-SC0022295. W.C. acknowledges support from the US National Science Foundation (DMR-2238204). This research used Electron Microscopy resources of the Center for Functional Nanomaterials (CFN), which is a U.S. Department of Energy Office of Science User Facility, at Brookhaven National Laboratory under Contract No. DE-SC0012704. The authors thank Prof. Andrew M. Minor and Dr. Colin Ophus from Lawrence Berkeley National Laboratory, as well as Prof. Jun Ding from Xi'an Jiaotong University, for the helpful discussions. The authors thank Dr. Leixin Miao from the Pennsylvania State University (PSU) for his help on the

atomic-resolution STEM-EDX experiment. Also, the authors thank Prof. Chenyang Lu from Xi'an Jiaotong University and Prof. Xing Wang from PSU for their advice on the flash polishing parameters.

## Author contributions

Y.Y. and P.C. conceived the project. Y.H. performed all TEM sample preparation and electron microscopy characterization. H.C. performed the MD simulations. Y.S. and Y.H. performed the electron microscopy data analysis. J.L. and W.C. performed the bulk sample fabrication and the XRD experiments. Y.Y. and S.W. fabricated the CoNiV bulk sample. Z.Z. performed the IM3D simulations. B.X. provided important feedback on the simulation data analysis. M.L. and J.Y. performed the in-situ TEM annealing experiment. Y.Y., P.C., Y.H., and H.C. wrote the manuscript. Y.H., H.C., and Y.Z. plotted the figures. All authors contributed to the discussion of the results.

## Competing interests

The authors declare no competing interests.
