## [Peer Review File · Nature Communications]

Ubiquitous short-range order in multi-principal element alloys

Parts of this Peer Review File have been redacted as indicated to maintain the confidentiality of unpublished data.Reviewers' comments:

Reviewer #1 (Remarks to the Author):

The authors characterized the development of short-range order (SRO) influenced by cooling rates and thermal processing in CoCrNi-based alloys using a novel electron microscopy technique. Further, the authors investigated the hypothesis that SRO forms during the solidification process at the solid/liquid interface using atomistic simulations. They reported that SRO arises directly from solidification, even under extreme cooling rates, and remains insensitive to thermal processing. The findings provide reasonable explanation for the variations in reported effects of SRO in FCC multi-principal element alloys as well as suggests new strategies to control material properties through the control of SRO through non-thermal means. There are some issues that need to be corrected and clarified before being accepted.

Comments:

1. In the Characterization section, it is asserted that the $\frac{1}{2}\{-311\}$ diffuse spots on the [112] ZA are considered direct evidence of SRO in FCC MPEAs citing [3,4]. However, a recent publication (<https://doi.org/10.1038/s41563-023-01570-9>) suggests that the same results from [3,4] may be a result of extra reflections from planar defects and that SRO "may not be definitely detected by the employed techniques". How can the authors be confident that the measurements obtained from EF-SAEF are indeed characterizing the presence of SRO and are not susceptible to the same issues potentially plaguing typical TEM work in previous literature discussed in (<https://doi.org/10.1038/s41563-023-01570-9>).
2. In Fig. 2f, it is shown in the upper panel that the Gaussian fit to the matrix peak "overshoots" the experimental data, something not observed in the lower panel for the SRO peak as it is more diffuse and not clipped. Is this phenomenon consistent throughout the fitting results for the matrix peaks in the manuscript used to produce the data for Fig. 2g and 2h? If so, how significant is this "overshoot" relative to the error bars already present in Fig. 2g and 2h?
3. What is the justification for the use of a modified Warren-Cowley parameter in Eqn. 1? It differs considerably from the typical expectation of the parameter established in the literature and is likely to be confusing at first glance by readers. Would the data presented in Figure 3 change considerably enough to affect the interpretation of the results by using the more common expression of the Warren-Cowley parameter?
4. How is the term Ni-j,0 computed in Eqn. 1? Is this an averaged term?
5. Why is the colorbar in Fig. 3f truncated at the lower end (" <2 ") when the distributions presented in Fig. 3g are much wider and this truncation would bias the color scale to the right tail of the distribution. The colorbar should reasonably capture zero in order to better highlight the RSS region. Further, the colorbar seems non-linear and the lower region (dark blue, " <2 " region) appears exaggerated.
6. In the section "Ion diffusivity in liquid" on line 605, "doesn't influence" should be "isn't influenced". Also, on line 606, what are the units for the "20x20x20 simulation box"? Is this length units, number of unit cells, etc?

Reviewer #2 (Remarks to the Author):

The present manuscript uses different techniques including additive manufacturing to produce CoCrNi, CoCrFeNi and CoCrFeMnNi. This allows them to study the short range order in these alloys over a wide range of cooling rates, showing no significant difference. They use extensive simulations to show that short range order is formed during the solidification process even at high cooling rate. The manuscript gives a nice overview of the literature in the field is well motivated and has an extensive discussion, but some of their conclusions require more convincing experimental work.

-Figure 2f shows the fitting quality check. The experimental profile clearly shows that the fundamental peak was overexposed making the relative SRO intensity erroneous. This needs to be checked and corrected for all measurements.

-The in situ nanomechanical sample (Fig 5c) looks very damaged and the initial relative SRO intensity does not match with the experiment. Therefore, it would be helpful to provide better TEM images to prove that the initial structure is indeed relevant.

Also, CoNiV is not shown in Figure 2g, making it difficult to assess how relevant the small change in relative intensity shown in Figure 5e actually is (it seems to be less than some differences in Fig 2g).

- Finally, it would be helpful to provide better TEM images for the different samples, to demonstrate chemical and microstructural homogeneity and exclude any differences between the samples. From the low magnification images given in Figure 11 it is very difficult to assess what the different contrast features are. Also, it is quite surprising that the dislocation density is higher after annealing at 900°C. Therefore, a better discussion of the microstructure should precede the SRO analysis.

Reviewer #3 (Remarks to the Author):

The authors examine the prevalence of short range order in CoCrNi and related higher-order MPEAs, using an "Enhanced Quantitative SAED" method. The alloys were prepared at 3 quench rates spanning ~ 7 orders of magnitude, and were also examined in the annealed condition. The authors convincingly demonstrate that SRO is present even in the 5 element MPEA prepared at the highest quench rate. Variations in compositional complexity and thermal preparation are shown to have very little impact on the relative degree of SRO, which is counter to most assumptions in the MPEA field. The authors go on to show that the degree of SRO can be changed via plastic deformation. This experimental work is quite well-presented and will have significant implications for the design of MPEAs.

Based on their experimental observations, the authors describe a local diffusion, solidification, and remelting model for the development of SRO during solidification, which they demonstrate using MD simulations. Their simulations show that the SRO develops very quickly in both space and time; the transition zone from random solid solution to solid with SRO occurs within only 6 atomic layers, and SRO is observed at solidification rates up to 10^{11} K/s. I found this model to be much less convincing than the experimental work. My first inclination would be to hypothesize that some level of chemical ordering develops in the liquid prior to crystallization. Although I am primarily an experimentalist, I wonder if the lack of ordering in the simulated supercooled liquid and the rapid ordering in the simulated solid could be the result of the EAM potentials chosen for the simulations. The authors should address how the potentials were selected and whether they were optimized for the simulations of solidification.

Related to this model, on page 13 the authors indicate that the Ni-Ni bonds would be favored over Ni-Co or Ni-Cr, leading to local melting of the latter arrangements until the Ni in the vicinity is depleted. However, the enthalpy of mixing of Ni-Cr in particular is negative (as is Co-Cr, while that for Ni-Co is 0). In view of this, why would the Ni-Ni bonds be favored? Overall, the authors need to better justify how the small differences in the (negative) enthalpies of mixing could result in such rapid development of SRO during solidification.

An itemized list of responses to reviewers' remarks
(Blue: Reviewer's comments; Black: Our response)

Table of Contents

Reviewer: #1	2
Reviewer #2	19
Reviewer #3	40
Reference	49

Reviewer: #1

Comment 1

The authors characterized the development of short-range order (SRO) influenced by cooling rates and thermal processing in CoCrNi-based alloys using a novel electron microscopy technique. Further, the authors investigated the hypothesis that SRO forms during the solidification process at the solid/liquid interface using atomistic simulations. They reported that SRO arises directly from solidification, even under extreme cooling rates, and remains insensitive to thermal processing. The findings provide reasonable explanation for the variations in reported effects of SRO in FCC multi-principal element alloys as well as suggests new strategies to control material properties through the control of SRO through non-thermal means. There are some issues that need to be corrected and clarified before being accepted.

Response:

We thank the reviewer for the encouraging and helpful comments, especially for pointing out that our electron microscopy technique is "novel" and that "*The findings provide a reasonable explanation for the variations in reported effects of SRO in FCC multi-principal element alloys as well as suggest new strategies to control material properties.*" We are happy to revise and address the issues the reviewer has raised.

Comment 2:

In the Characterization section, it is asserted that the $\frac{1}{2}\{-311\}$ diffuse spots on the [112] ZA are considered direct evidence of SRO in FCC MPEAs citing [3,4]. However, a recent publication (<https://doi.org/10.1038/s41563-023-01570-9>) suggests that the same results from [3,4] may be a result of extra reflections from planar defects and that SRO "may not be definitely detected by the employed techniques". How can the authors be confident that the measurements obtained from EF-SAEF are indeed characterizing the presence of SRO and are not susceptible to the same issues potentially plaguing typical TEM work in previous literature discussed in (<https://doi.org/10.1038/s41563-023-01570-9>).

Response:

Thank you for your insightful comments. We understand the concern regarding the direct imaging of chemical short-range order (CSRO), which indeed presents a grand challenge. Previous theoretical studies¹ have suggested that SRO is a distinct feature of high-entropy alloys (HEAs) and multi-principal element alloys (MEAs) compared to traditional alloys. However, the experimental characterization of SRO in MEAs, especially in relation to deformation, remains elusive due to complexities in characterization.

Recent publications in journals such as *Nature* have attempted to provide direct evidence of SRO in MEAs through methods like diffuse diffraction signals found in selected area electron diffraction (SAED) patterns^{2,3}. For example, the streaks along $\{111\}$ direction on the [110] zone axis², and the Bragg spots on the $\frac{1}{2}\{-311\}$ location on the [112] zone axis have been indicated as evidence of SRO in MEAs³. However, these methods have been met with contention. As you pointed out, a recent comment in *Nature Materials* questions the origin of the diffuse signals and

suggests they may arise from planar defects rather than SRO⁴. In addition, a recent research article⁵ in *Nature* considered these diffuse signals as an outcome of reflections from higher-order Laue zones (HOLZs) rather than SRO. These discussions have indeed added complexity to the interpretation of TEM results.

However, none of these papers definitively prove that SRO cannot lead to diffuse signals in the $\frac{1}{2}\{-311\}$ location on the $[112]$ zone axis. They primarily highlight that other factors can cause similar signals and that certain proposed SRO structures may not generate signals in this location. The critical gap lies in the uncertainty of the SRO structure in MEAs and whether it can produce the diffuse signals in question. In other systems like HCP and BCC, diffuse spots have been widely used as indicators of SRO⁶. For instance, a recent paper⁷ used diffuse spots in SAED patterns to characterize SRO in BCC HEAs (Fig. R1). Notably, the absence of stacking faults in BCC materials strengthens the likelihood that the observed diffuse spots are attributable to SRO rather than to stacking faults.

Fig. R1. a. Characterization of SRO in BCC HEAs based on the diffuse spots in SAEDs. Figure reproduced from Ref⁷.

In our view, it is likely that SRO, SF, and HOLZ all contribute to the diffuse signals in the $\frac{1}{2}\{-311\}$ location on the $[112]$ zone axis. The key is to decouple these effects. We believe that dynamic imaging, as employed in our in-situ TEM experiments (see Fig. R2 below and Fig. 5 in the manuscript), where we perform nano-mechanical testing, is a more effective approach to distinguish these effects. Specifically, during cyclic loading, the emission and movement of dislocations from the crack tip are expected to disrupt SRO⁸, leading to a decrease in its degree as the number of cycles increases. This hypothesis is supported by our observations: we noted a reduction in the diffuse signal in the relevant region correlating with an increase in cycle numbers, strongly suggesting a predominant role of SRO on the diffuse signals (Fig. R2).

Fig. R2. In-situ TEM experiments to study the evolution of SRO during cyclic loading at a crack tip. This image is an updated image of Figure 5 in our manuscript.

Our observation of the evolution of SRO in these in-situ TEM experiments also aligns with a previous work using the electric resistivity method to measure changes in SRO during deformation (**Fig. R3** below).

Fig. R3. Variations of electrical resistivity for single crystals water-quenched from 1473 K and subsequently annealed at 673 K for 504 h measured as a function of plastic strain in stage I in tensile deformation. When the strain is low, a reduction of the electric resistivity was observed, indicating a reduction of SRO. The figure is modified and reproduced from Ref. ⁹.

In contrast, for stacking faults (SFs), an increase rather than a decrease would be anticipated during cyclic loading as a result of the accumulation of plastic deformation. This is exemplified in **Fig. R4**, which depicts the deformation morphology of a CrCoNi MEA after fatigue loading, characterized by abundant SFs/Twins. Also, **Fig. R5** shows the molecular dynamics (MD) simulation of pure copper, which exhibits a face-centered cubic (FCC) structure and a low SF energy, analogous to those of the CrCoNi medium entropy alloy. Profuse stacking faults will develop during cyclic loading near the crack tip.

Figure R4. TEM image showing the microstructure after fatigue fracture of coarse grain structure CrCoNi MEA. Ref. ¹⁰.

Figure R5. Snapshots showing the plastic deformation and crack growth during fatigue loading for a single crystal pure Cu. (a) 2nd cycle; (b) 3rd cycle; (c) 4th cycle; (d) 8th cycle; (e) 9th cycle; (f) 11th cycle. Figure reproduced from Ref. ¹¹.

In terms of HOLZ effects, should the diffuse signals originate from HOLZ, we would anticipate no significant changes during the cyclic loading of MEA, given that neither the FCC structure nor the sample thickness undergoes alteration.

Based on the analysis above, we believe our dynamical imaging (**Fig. 5** in the manuscript) provides compelling evidence that SRO is the principal contributor to the diffuse spots at the $\frac{1}{2}\{-311\}$ location on the $[112]$ zone axis in MEAs.

Lastly, we want to emphasize that our observation of similar amounts of SRO in both annealed and additive manufactured (ultrafast quenched) samples corroborate previous studies using methods like high-energy synchrotron X-ray diffraction¹² (**Fig. R6** below). This further validates the reliability of our electron microscopy characterization of SRO.

We have added discussions in the main text to further address this concern. The traced changes are displayed below in **Figs. R7-R8**.

Fig. R6. High-energy synchrotron X-ray diffraction¹² study showing that the distribution of different types of CSRO is similar for the annealed sample and the SLMed sample (additive manufactured with a high cooling rate). Figure is reproduced from Ref. ¹².

Thin lamellas for TEM characterization were prepared from the bulk samples, with their bright-field TEM images and corresponding SAED patterns shown in **Fig. 1b**. All the characterizations were performed with the same instrument and alignment conditions to ensure the robustness of the results. In prior research, diffuse spots at the $\frac{1}{2} \{311\}$ location of FCC MPEAs electron diffraction on $[112]$ zone axis has been considered as direct evidence of SRO^{3,4} (see more discussions about differentiating the effects of SRO, planar defects and higher-order Laue zones (HOLZ) on the diffuse signals in the section for **Fig. 5**). Remarkably, the discernible existence of these diffuse spots in the SAED patterns in **Fig. 1b** reveals that all ten samples exhibit SRO, even the LPBF sample fabricated at the cooling rate of 10^7 K s^{-1} . A schematic time-temperature-transformation (TTT) diagram is plotted in **Fig. 1c** to facilitate the visualization of cooling rate, thermal processing history and the associated degree of SRO.

Fig. R7. Update of the manuscript on page 5.

Mechanical tuning of SRO

To investigate additional methods that could be effectively integrated with thermal treatment and quench rate selection for enhanced SRO tuning, we have conducted in-situ nano-mechanical experiments using the EQ-SAED method. This allows us to elucidate the impact of mechanical deformation and dislocation slip on SRO. CoNiV was selected for this demonstration as it offers a better SNR in its SRO signal (**Fig. 2h**), which enables more precise quantification of the mechanical deformation effect. We fabricated a push-to-pull (PTP) setup from bulk CoNiV using a focused ion beam (FIB), as depicted in **Fig. 5 a,b**. This device offers a straightforward implementation of the tensile testing of a thin sample within the limited space of a TEM column, while ensuring stability throughout the loading process. **Figure 5c** illustrates that the EQ-SAED characterization was performed on the region circled ahead of the prefabricated crack tip at which the deformation concentrates. The loading curve is presented in **Fig. 5d**. It is important to note that the entire electron transparent region maintained a single-crystal structure without planar defects during the experiments. Previous studies have noted that SRO can be disrupted by dislocations gliding across it^{5,9,22}. In the case of cyclic loading, dislocations are repetitively emitted from the crack tip and migrate away from it during loading, and some of the dislocations are fully and partially recovered in the subsequent unloading. The net positive dislocation emission and gliding in each cycle allow us to study its effect on SRO. Aligning with this theoretical speculation, we observed a consistent decrease in relative SRO intensity as the number of cycles increased (**Fig. 5e**). A comparison of all the samples in this study is shown in **Supplementary Fig. 29**. Compared to thermal treatment and changing the cooling rate, mechanical tuning of SRO is more controllable.

Moreover, our experiment confirms the ability of the EQ-SAED method to capture SRO information in MPEAs quantitatively, further reinforcing its effectiveness and reliability. Recent studies have proposed that these diffuse signals at the $\frac{1}{2} \{311\}$ location of FCC MPEAs electron diffraction on [112] zone axis (a key aspect of our EQ-SAED approach) have alternative origins such as stacking faults⁴⁸ or HOLZ⁴⁹. In the case of stacking faults, we would expect an increase in diffuse signals during cyclic loading, due to the accumulation of plastic deformation. Conversely, if these signals were primarily a result of HOLZ, no significant changes should be observed during the cyclic loading of MEAs. Only the decrease in the SRO due to mechanical deformation aligns with our findings that the relative intensity of diffuse signals reduces during cyclic loading. Therefore, our in-situ transmission electron microscopy (TEM) experiments provide robust evidence of a strong correlation between the diffuse spots on the [112] zone axis and the degree of SRO in the material.

Fig. R8. Update of the manuscript on page 15.

Comment 3:

In Fig. 2f, it is shown in the upper panel that the Gaussian fit to the matrix peak “overshoots” the experimental data, something not observed in the lower panel for the SRO peak as it is more diffuse and not clipped. Is this phenomenon consistent throughout the fitting results for the matrix peaks in the manuscript used to produce the data for Fig. 2g and 2h? If so, how significant is this “overshoot” relative to the error bars already present in Fig. 2g and 2h?

Response:

Thank you for bringing attention to this aspect in **Fig. 2f**. We realize that our previous version of the manuscript did not sufficiently explain this phenomenon. The ‘overshoot’ you observed in the Gaussian fit to the matrix peak is due to detector saturation, or overexposure. When the intensity surpasses the upper limit of the detector, it results in a flat line in the data.

In the initial version of our manuscript, we intentionally showed the overexposed image to highlight the advantages of our method, *i.e.*, this technique negates the need for costly TEM equipment while ensuring data reliability by innovatively extracting signals from overexposed data. However, we inadvertently omitted a discussion on this topic. The fact that we openly presented these ‘imperfect’ data underscores our confidence in our approach.

However, the true signal of the matrix peak should be Gaussian rather than the truncated Gaussian curve with a plateau (shown by the blue curve in Fig. 2f). This can be proved by an image captured using a much lower current with the same exposure time or the same current but with a shorter exposure time, avoiding overexposure. Note that the beam current can be controlled by the strength of the condenser-2 lens (C2). The relationship between beam current and C2 strength for our TEM is also measured and shown in **Fig. R9** below. Also, **Fig. R10** shows the brightest matrix peak in an image taken with a shorter exposure time (100 ms), where the matrix peak is not overexposed and shows a nice Gaussian shape.

Fig. R9. Our measurement of the relationship between C2 and electron beam current. Data was collected at a $22,500\times$ magnification without a select area aperture inserted.

Fig. R10. Matrix peak without overexposure shows a perfect Gaussian shape agreeing with our fitting result. The C2 strength is 49.767%.

In our experiment, we deliberately chose a beam current and an exposure time that overexposed the brightest matrix peak for several reasons:

1. The C2 aperture (beam current) was selected to approximate parallel beam conditions, ensuring sharper and more reliable SRO peaks.
2. Extracting a reliable SRO signal is challenging due to its weak intensity and diffuse nature. When the exposure time is short or the beam current is low, the intensity of the SRO peak is comparable to the noise level, and the reliability of the characterization will be compromised. A higher current or longer exposure time was necessary to ensure a sufficient signal-to-noise ratio for the SRO peak. Ideally, an energy filter coupled with a direct electron detector could enhance the signal-to-noise ratio at a low electron beam current or exposure time, but such equipment was costly and not available in our facility.
3. Our method can recover the actual height of the matrix peak by fitting the data outside the overexposed region (**Fig. R11**). This approach is both robust and innovative, as it negates the need for the expensive equipment mentioned earlier.

Fig. R11. An image illustrating the overexposed region and the data range for fitting.

Please note that before our experiments, we systematically studied how the currents will affect the SRO detection quality. **Fig. R12** shows that the error of SRO relative intensity increases as the C2 strength increases (*i.e.*, electron beam current reduces) while the exposure time is fixed. Based on this study, we chose a C2 strength, which is close to the parallel beam condition and showed a small error.

Fig. R12. The change of error bar for SRO relative intensity as a function of C2.

To clarify this, we have now added comprehensive discussions in both the manuscript and supplementary materials, as illustrated in **Figs. R13-R15**.

demonstrates the robustness of the method. Note that the selection of the beam current and exposure time in our experiments was a strategic decision aimed at enhancing the signal-to-noise ratio for the SRO signal and ensuring a close-parallel-beam condition. This choice inevitably led to the overexposure of the three brightest matrix peaks, as observed in our results. However, our analysis method meticulously avoids these overexposed regions during the fitting process, thereby ensuring the accuracy of our data. More detailed discussion can be found in **Supplementary Notes A** and **Supplementary Fig. 14-17**. At last, the SRO Bragg peak intensity divided by the matrix Bragg peak intensity, denoted as the “relative SRO intensity”, is computed to compare different MPEA samples with different compositions and thermal treatments (**Fig. 2 g-h**).

Fig. R13. Update of the manuscript on page 7.

Supplementary Notes

A. Overexposure of the matrix peaks for better SRO detection

In Fig. 2f, the fitting curves seem to have an “overshoot” compared with the experimental data. The ‘overshoot’ is due to detector saturation or overexposure. When the intensity surpasses the upper limit of the detector, it results in a flat line in the data. The relationship between beam current and C2 strength for our TEM is measured and shown in **Supplementary Fig. 14**. Also, **Supplementary Fig. 15** shows the brightest matrix peak in an image without overexposure, where the matrix peak shows a nice Gaussian shape.

In our experiment, we deliberately chose a beam current and an exposure time that overexposed the brightest matrix peak for several reasons:

1. The C2 aperture (beam current) was selected to approximate parallel beam conditions, ensuring sharper and more reliable SRO peaks.
2. Extracting a reliable SRO signal is challenging due to its weak intensity and diffuse nature. When the exposure time is short or the beam current is low, the intensity of the SRO peak is comparable to the noise level, and the reliability of the characterization will be compromised. A higher current or longer exposure time was necessary to ensure a sufficient signal-to-noise ratio for the SRO peak. Ideally, an energy filter coupled with a direct electron detector could enhance the signal-to-noise ratio at a low electron beam current or exposure time, but such equipment was costly and not available in our facility.
3. Our method can recover the actual height of the matrix peak by fitting the data outside the overexposed region (**Supplementary Fig. 16**). This approach is robust and it negates the need for the expensive equipment mentioned earlier.

Please note that before our experiments, we systematically studied how the currents will affect the SRO detection quality. **Supplementary Fig. 17** shows that the error for the relative intensity of SRO increases as the C2 strength increases (i.e., electron beam current reduces). We chose a C2 strength of 49.767%, which is close to the parallel beam condition and shows a small error bar.

Fig. R14. Update of the supplementary note.

Supplementary Fig. 14 | Measurement of the relationship between C2 and electron beam current. Data was collected at 22,500 × magnification without a select area aperture inserted.

Supplementary Fig. 15 | Matrix peak without overexposure shows a perfect Gaussian shape agreeing with our fitting result. The C2 strength is 49.767%.

Supplementary Fig. 16 | An image illustrating the overexposed region and the data range for fitting.

Supplementary Fig. 17 | The change of error bar for SRO relative intensity as a function of C2.

Fig. R15. Update of the supplementary figures.

Comment 4:

What is the justification for the use of a modified Warren-Cowley parameter in Eqn. 1? It differs considerably from the typical expectation of the parameter established in the literature and is likely to be confusing at first glance by readers. Would the data presented in Figure 3 change considerably enough to affect the interpretation of the results by using the more common expression of the Warren-Cowley parameter?

Response.

The modified order parameter used in this manuscript is $\Delta\alpha_{i,j} = N_{i,j} - N_{i,j,0}$, where $N_{i,j}$ is the actual number of i - j pairs in the first nearest neighbour and $N_{i,j,0}$ is the number of i - j pairs for the random solid solution. A positive $\Delta\alpha_{i,j}$ indicates a favoured and increased number of pairs, meaning that element i tends to bond with element j , while a negative value represents an unfavoured pairing. Therefore, the value of $\Delta\alpha_{i,j}$ directly quantifies the number of i - j pairs changed from random mixing. For example, $\Delta\alpha_{Ni-Ni} = 2$ indicates there are two extra Ni-Ni pairs when comparing with a random solid solution.

Following the reviewer's suggestion, we also computed the more well-known Warren-Cowley parameter for comparison. The Warren-Cowley parameter for multicomponent systems in the following formula^{8,13,14}:

$$\alpha_m^{ij} = \frac{P_m^{ij} - c_j}{\delta^{ij} - c_j}$$

where m means the m^{th} nearest neighbouring shell, P_m^{ij} is the probability of finding a j -type atom around the i -type atom in the m^{th} shell, δ^{ij} is the Kronecker delta function. It is noted that for the same type, $i=j$ (i.e., $\delta^{ij}=1$), a positive α_m^{ij} suggests the tendency of segregation. When $i \neq j$ ($\delta^{ij}=0$), a negative α_m^{ij} suggests the tendency of segregation.

In **Fig. R16**, we show the comparison of these two kinds of order parameters, namely modified parameter $\Delta\alpha_{i,j}$ and Warren-Cowley order parameter $\alpha_{m=1}^{ij}$. As can be seen, the spatial distributions of order parameters from the two calculations exhibit essentially the same characteristics. It is worth noting that the Warren-Cowley order parameters, ranging from -2 to 1, can be positive (Ni-Ni) or negative (Co-Cr) even for the same ordering (segregation). The results indicate that the parameter $\Delta\alpha_{i,j}$, carrying the essential feature of the Warren-Cowley parameter, more straightforwardly reflects the degree of local chemical ordering.

To avoid possible confusion, we have added the Warren-Cowley parameter and extended the discussion in the revised manuscript and supplementary materials (illustrated in **Figs. R17-R18** and **Supplementary Fig. 18**).

Fig. R16 (a) Pairwise order parameter calculated from modified Warren-Cowley parameter, $\Delta\alpha_{i-j}$. (b) Pairwise order parameter calculated from Warren-Cowley $\alpha_{m=1}^{ij}$. (c) Probability distribution of α_1^{Ni-Ni} and α_1^{Co-Cr} .

Local chemical short-range order parameter The local short-range order was calculated as the nonproportional number of local atomic pairs $\Delta\alpha_{i-j}$, modified from the Warren-Cowley parameter^{1,63,64}:

$$\Delta\alpha_{i-j} = N_{i-j} - N_{i-j,0} \quad (1)$$

where N_{i-j} is the number of $i-j$ pairs in the first nearest neighbor shell of an i -type atom, $N_{i-j,0}$ is the number of $i-j$ pairs for the random solid solution. Since there are 12 first nearest neighboring atoms for FCC structure, an equimolar CoCrNi would give rise to $N_{i-j,0} = 4$ (i.e., $12 \times 1/3 = 4$). For species i and j , a positive $\Delta\alpha_{i-j}$ suggests the clustering in the first shell and the negative $\Delta\alpha_{i-j}$ means opposite. The difference between this definition of SRO parameters and the more commonly used Warren-Cowley parameters are provided in **Supplementary Notes B**, showing that the definition of SRO parameter will not change the conclusion and reliability of this work.

Fig. R17. Update of the method section.

B. CSRO parameter

The modified order parameter used in this manuscript is $\Delta\alpha_{i-j} = N_{i-j} - N_{i-j,0}$, where N_{i-j} is the actual number of $i-j$ pairs in the first nearest neighbour and $N_{i-j,0}$ is the number of $i-j$ pairs for the random solid solution. A positive $\Delta\alpha_{i-j}$ indicates a favoured and increased number of pairs, meaning that element i tends to bond with element j , while a negative value represents an unfavoured pairing. Therefore, the value of $\Delta\alpha_{i-j}$ directly quantifies the number of $i-j$ pairs changed from random mixing. For example, $\Delta\alpha_{Ni-Ni} = 2$ indicates there are two extra Ni-Ni pairs when comparing with random solid solution.

We have also computed the more well-known Warren-Cowley parameter for comparison. The Warren-Cowley parameter for multicomponent systems in the following formula¹⁻³:

$$\alpha_m^{ij} = \frac{P_m^{ij} - c_j}{\delta^{ij} - c_j}$$

where m means the m^{th} nearest neighbouring shell, P_m^{ij} is the probability of finding a j -type atom around the i -type atom in the m^{th} shell, δ^{ij} is the Kronecker delta function. It is noted that for the same type, $i=j$ (i.e., $\delta^{ij}=1$), a positive α_m^{ij} suggests the tendency of segregation. When $i \neq j$ ($\delta^{ij}=0$), a negative α_m^{ij} suggests the tendency of segregation.

In **Supplementary Fig. 18**, we show the comparison of these two kinds of order parameters, namely modified parameter $\Delta\alpha_{i-j}$ and Warren-Cowley order parameter $\alpha_{m=1}^{ij}$. As can be seen, the spatial distributions of order parameters from the two calculations exhibit essentially the same characteristics. It is worth noting that the Warren-Cowley order parameters, ranging from -2 to 1, can be positive (Ni-Ni) or negative (Co-Cr) even for the same ordering (segregation). The results indicate that the parameter $\Delta\alpha_{i-j}$, carrying the essential feature of the Warren-Cowley parameter, more straightforwardly reflects the degree of local chemical ordering.

Fig. R18. Update of the supplementary materials.

Comment 5:

How is the term $N_{i-j,0}$ computed in Eqn. 1? Is this an averaged term?

Response.

$N_{i-j,0}$ is the number of $i-j$ pairs in the system of random solid solution. Since there are 12 first nearest neighboring atoms for FCC structure, an equimolar CoCrNi would give rise to $N_{i-j,0} = 4$ (i.e., $12 \cdot 1/3 = 4$). The values therefore represent system-averaged. We have added this info in our method section.

Comment 6:

Why is the colorbar in Fig. 3f truncated at the lower end (“<2”) when the distributions presented in Fig. 3g are much wider and this truncation would bias the color scale to the right tail of the distribution. The colorbar should reasonably capture zero in order to better highlight the RSS region. Further, the colorbar seems non-linear and the lower region (dark blue, “<2” region) appears exaggerated.

Response.

We want to thank the reviewer for the careful and helpful insights.

We used truncation/threshold of colorbar to enhance the contrast and highlight the regimes with a high degree of chemical short-range order (i.e., >2). For clarity, we present in **Fig. R19a** the spatial distribution of the order parameters, $\Delta\alpha_{\text{Ni-Ni}}$ and $\Delta\alpha_{\text{Co-Cr}}$ using the unmodified colorbar. Because Ni-Ni and Co-Cr are favored pairings, positive value implies the degree of such ordering. The truncation value of 2 extracts all the atoms having more than two favored pairs. The atoms with values smaller than two are colored by dark blue. **Fig. R19b** shows the distribution after the truncation, which results in a non-linear color bar. The threshold (truncation of the color bar) does not change our major conclusions, but it enhances the image contrast to help the readers capture the difference more easily.

We have updated the figure caption and color bar in the amended manuscript to clarify this. The result without truncation/threshold (**Fig. R20**) is also included in the **Supplementary Fig. 19**.

Fig. R19. The spatial distribution of pairwise order parameters $\Delta\alpha_{\text{Ni-Ni}}$ and $\Delta\alpha_{\text{Co-Cr}}$. (a) The original colorbar, which ranges from -4 to 7. (b) The colorbar with a threshold. Atoms with order parameters smaller than 2 are colored by dark blue.

Fig. 3 | Modeling the evolution of chemical distribution during solidification and after annealing. a,b, Snapshots showing the solidification process. Green represents the FCC crystal structure, red represents the solidification front, and purple represents the liquid. c,d, Atomic slices of the structure (in the x - z plane) showing the distribution of atoms (c) and the pairwise order parameters $\Delta\alpha_{\text{Ni-Ni}}$ and $\Delta\alpha_{\text{Co-Cr}}$ after solidification (d), respectively. A threshold of 2 has been applied on the colorbar to enhance the contrast. Thus, the atoms with order parameters smaller than 2 are colored by dark blue. The original result without threshold is included in **Supplementary Fig. 19**. e,f, Atomic slices of the structure (in the x - z plane) showing the distribution of atoms and

Fig. R20. Update of the Figure 3.

Comment 7:

In the section “Ion diffusivity in liquid” on line 605, “doesn’t influence” should be “isn’t influenced”. Also, on line 606, what are the units for the “20x20x20 simulation box”? Is this length units, number of unit cells, etc?

Response.

We would like to thank the reviewer very much for the careful examination. The unit in “20×20×20 simulation box” is unit cell. We have corrected the typos and updated the units as “We created a system with a dimension of $20a_0 \times 20a_0 \times 20a_0$ (a_0 is lattice constant) that contained 32,000 atoms.”, as shown in **Fig. R21**.

Ion diffusivity in liquid

The diffusion of elements in the liquid is characterized by diffusion coefficient D via the Einstein-Smoluchowski equation^{65,66},

$$D = \frac{\langle \mathbf{R}^2 \rangle}{6dt} \quad (2)$$

where d is the dimension of diffusion and is equal to 3 in our simulations. t is the time and $\langle \mathbf{R}^2 \rangle$ is the mean square displacement (MSD) of an ensemble of particles over time that can be written as follows:

$$\langle \mathbf{R}^2 \rangle = \frac{1}{N} \sum_{i=1}^N |\mathbf{r}_i(t) - \mathbf{r}_i(0)|^2 \quad (3)$$

where N is the number of each element in the liquid, $\mathbf{r}_i(t)$ is the position of atom i at time t and $\mathbf{r}_i(0)$ is the initial position of atom i . We performed independent simulations to ensure that the measurement of diffusion at solidification temperatures isn't influenced by solidification. We created a system with a dimension of $20a_0 \times 20a_0 \times 20a_0$ (a_0 is lattice constant) that contained 32,000 atoms. Periodic boundary conditions were applied to three dimensions. The system's temperature was increased to 2,000 K for 10 ps to melt the crystal completely and then relaxed at target temperature for 10 ps. Afterward, the MSD of each element was calculated by Equation. 3 every 1 ps. The square root of diffusivity measures the maximum diffusion distance three-dimensionally in the unit of time. Diffusion distance is calculated by:

$$r = \sqrt{Dt} \quad (4)$$

where D is the diffusivity and t is the time.

Fig. R21. Update of the method section.

Reviewer #2

Comment 1:

The present manuscript uses different techniques including additive manufacturing to produce CoCrNi, CoCrFeNi and CoCrFeMnNi. This allows them to study the short range order in these alloys over a wide range of cooling rates, showing no significant difference. They use extensive simulations to show that short range order is formed during the solidification process even at high cooling rate. The manuscript gives a nice overview of the literature in the field is well motivated and has an extensive discussion, but some of their conclusions require more convincing experimental work.

Response:

We appreciate reviewer #2 for the helpful comments and encouraging feedback, especially for pointing out that our work “*use extensive simulations to show that short range order is formed during the solidification process even at high cooling rate*” and “*The manuscript gives a nice overview of the literature in the field is well motivated and has an extensive discussion*”.

We have addressed the concerns of reviewer #2 with a more detailed discussion and additional extensive experimental data, which will be presented below.

Comment 2:

Figure 2f shows the fitting quality check. The experimental profile clearly shows that the fundamental peak was overexposed making the relative SRO intensity erroneous. This needs to be checked and corrected for all measurements.

Response:

Thank you for bringing attention to this aspect in **Fig. 2f**. We have addressed the same concern from reviewer #1 above. Here we just replicate what we have shown above to facilitate the reading process.

The 'overshoot' you observed in the Gaussian fit to the matrix peak is due to detector saturation, or overexposure. When the intensity surpasses the upper limit of the detector, it results in a flat line in the data.

However, the true signal of the matrix peak should be Gaussian rather than the truncated Gaussian curve with a plateau (shown by the blue curve in Fig. 2f). This can be proved by an image captured using a much lower current with the same exposure time or the same current but with a shorter exposure time, avoiding overexposure. Note that the beam current can be controlled by the strength of the condenser-2 lens (C2). The relationship between beam current and C2 strength for our TEM is also measured and shown in **Fig. R9** below. Also, **Fig. R10** shows the brightest matrix peak in an image taken with a shorter exposure time (100 ms), where the matrix peak is not overexposed and shows a nice Gaussian shape.

Fig. R9. Our measurement of the relationship between C2 and electron beam current. Data was collected at a 22,500 × magnification without a select area aperture inserted.

Fig. R10. Matrix peak without overexposure shows a perfect Gaussian shape agreeing with our fitting result. The C2 strength is 49.767%.

In our experiment, we deliberately chose a beam current and an exposure time that overexposed the brightest matrix peak for several reasons:

1. The C2 aperture (beam current) was selected to approximate parallel beam conditions, ensuring sharper and more reliable SRO peaks.
2. Extracting a reliable SRO signal is challenging due to its weak intensity and diffuse nature. When the exposure time is short or the beam current is low, the intensity of the SRO peak is comparable to the noise level, and the reliability of the characterization will be compromised. A higher current or longer exposure time was necessary to ensure a sufficient signal-to-noise ratio for the SRO peak. Ideally, an energy filter coupled with a direct electron detector could enhance the signal-to-noise ratio at a low electron beam current or exposure time, but such equipment was costly and not available in our facility.

- Our method can recover the actual height of the matrix peak by fitting the data outside the overexposed region (**Fig. R11**). This approach is both robust and innovative, as it negates the need for the expensive equipment mentioned earlier.

Fig. R11. An image illustrating the overexposed region and the data range for fitting.

Please note that before our experiments, we systematically studied how the currents will affect the SRO detection quality. **Fig. R12** shows that the error of SRO relative intensity increases as the C2 strength increases (*i.e.*, electron beam current reduces) while the exposure time is fixed. Based on this study, we chose a C2 strength, which is close to the parallel beam condition and showed a small error.

Fig. R12. The change of error bar for SRO relative intensity as a function of C2.

To clarify this, we have now added comprehensive discussions in both the manuscript and supplementary materials, as illustrated in **Figs. R13-R15**.

demonstrates the robustness of the method. Note that the selection of the beam current and exposure time in our experiments was a strategic decision aimed at enhancing the signal-to-noise ratio for the SRO signal and ensuring a close-parallel-beam condition. This choice inevitably led to the overexposure of the three brightest matrix peaks, as observed in our results. However, our analysis method meticulously avoids these overexposed regions during the fitting process, thereby ensuring the accuracy of our data. More detailed discussion can be found in **Supplementary Notes A** and **Supplementary Fig. 14-17**. At last, the SRO Bragg peak intensity divided by the matrix Bragg peak intensity, denoted as the “relative SRO intensity”, is computed to compare different MPEA samples with different compositions and thermal treatments (**Fig. 2 g-h**).

Fig. R13. Update of the manuscript on page 7.

Supplementary Notes

A. Overexposure of the matrix peaks for better SRO detection

In **Fig. 2f**, the fitting curves seem to have an “overshoot” compared with the experimental data. The ‘overshoot’ is due to detector saturation or overexposure. When the intensity surpasses the upper limit of the detector, it results in a flat line in the data. The relationship between beam current and C2 strength for our TEM is measured and shown in **Supplementary Fig. 14**. Also, **Supplementary Fig. 15** shows the brightest matrix peak in an image without overexposure, where the matrix peak shows a nice Gaussian shape.

In our experiment, we deliberately chose a beam current and an exposure time that overexposed the brightest matrix peak for several reasons:

1. The C2 aperture (beam current) was selected to approximate parallel beam conditions, ensuring sharper and more reliable SRO peaks.
2. Extracting a reliable SRO signal is challenging due to its weak intensity and diffuse nature. When the exposure time is short or the beam current is low, the intensity of the SRO peak is comparable to the noise level, and the reliability of the characterization will be compromised. A higher current or longer exposure time was necessary to ensure a sufficient signal-to-noise ratio for the SRO peak. Ideally, an energy filter coupled with a direct electron detector could enhance the signal-to-noise ratio at a low electron beam current or exposure time, but such equipment was costly and not available in our facility.
3. Our method can recover the actual height of the matrix peak by fitting the data outside the overexposed region (**Supplementary Fig. 16**). This approach is robust and it negates the need for the expensive equipment mentioned earlier.

Please note that before our experiments, we systematically studied how the currents will affect the SRO detection quality. **Supplementary Fig. 17** shows that the error for the relative intensity of SRO increases as the C2 strength increases (i.e., electron beam current reduces). We chose a C2 strength of 49.767%, which is close to the parallel beam condition and shows a small error bar.

Fig. R14. Update of the supplementary note.

Supplementary Fig. 14 | Measurement of the relationship between C2 and electron beam current. Data was collected at $22,500\times$ magnification without a select area aperture inserted.

Supplementary Fig. 15 | Matrix peak without overexposure shows a perfect Gaussian shape agreeing with our fitting result. The C2 strength is 49.767%.

Supplementary Fig. 16 | An image illustrating the overexposed region and the data range for fitting.

Supplementary Fig. 17 | The change of error bar for SRO relative intensity as a function of C2.

Fig. R15. Update of the supplementary figures.

Comment 3:

The in situ nanomechanical sample (Fig 5c) looks very damaged and the initial relative SRO intensity does not match with the experiment. Therefore, it would be helpful to provide better TEM images to prove that the initial structure is indeed relevant.

Response:

Thank you for highlighting the ambiguity in **Fig. 5c**. The original TEM image of **Figure 5c** was captured without an objective aperture inserted, resulting in poor contrast, making it challenging to conclusively determine if the sample is damaged (refer to **Fig. R22** below).

Fig. R22. The Fig. 5 in our initial manuscript. The sample ID is PTP#1.

In **Fig. R23a**, we showcase the same sample imaged with an objective aperture. Here, it is evident that the electron beam-transparent portion of the sample is a single crystal. The black line near the crack tip, marked in the figure, is a bending contour, not an indication of damage. Bending contours are typical in crystalline TEM samples, as the sample is seldom perfectly flat. A reliable way to verify a bending contour is to observe its movement during deformation/tilting. If it moves smoothly back and forth, it is likely a bending contour. In **Fig. R23**, snapshots taken during the in-situ TEM cyclic loading process confirm that the black line is a bending contour rather than a defect or damage.

Fig. R23. TEM snapshots showing the movement of bending contour during the cyclic deformation process in sample PTP#1. The white, blue, and yellow lines highlight the locations of the bending contour in a, b, and c, respectively.

From **Fig. R23**, while the sample (*Sample ID: PTP#1*) appears clean and devoid of significant defects like voids, cracks, or large planar defects/dislocations, it might still contain small dislocation loops too minuscule to discern at this magnification.

As our initial intent was to examine the evolution of the SRO signal in diffraction patterns, we did not capture higher-resolution images initially. Post the initial experiments, the sample was fractured. Therefore, it is not possible to get a high-resolution image showing the structure of the sample *PTP#1* before deformation. To validate our concept further, we have replicated the experiment with a new sample (*The sample ID is PTP#2*).

Fig. R24b,c,e displays results from this new sample (*PTP#2*), mirroring the conclusions of our previous **Fig. 5**: the sample is also a single crystal, and the SRO decreases with increasing deformation cycles.

Fig. R24. New Fig. 5 for the revised manuscript. The data is from the new sample (PTP#2).

Fig. R25 presents a higher resolution image of the sample pre-loading. Notably, tiny black/diffuse dots, indicative of dislocation loops from the focused ion beam (FIB) fabrication process, are visible.

Fig. R25. Enlarged image of Fig. R24c. (PTP#2).

It is well-acknowledged that FIB samples inevitably contain dislocation loops on the surface due to Ga ion irradiation^{15,16}. In this work, we employed two methods for TEM sample preparation:

1. **FIB + Flash Polishing:** After FIB fabrication, we used a flash polishing method (detailed in our methods section) to remove the FIB-induced surface damage. The samples in **Figures 1 and 2** were prepared using this approach.
2. **FIB Only:** This method was utilized for preparing the sample for in-situ TEM mechanical testing shown in **Fig. 5**. Flash polishing was not feasible here as it might compromise the stability of the mechanical testing process by etching the push-to-pull (PTP) structure or altering the crack shape, which is undesirable. The crack was deliberately created to localize the damage zone for a more precise capture of the SRO-deformation relationship.

Consequently, the difference in SRO relative intensity between **Figs. 1 and 5** stems from the distinct TEM sample fabrication methods. The FIB-only sample (**Fig. 5** in the manuscript) exhibits a weaker SRO signal than the flash-polished sample (**Fig. 1-2** in the manuscript), as the irradiation process is known to disrupt SRO¹⁷. Our finding also highlights that irradiation can be an effective method to tune SRO, which is also mentioned in our discussion in the manuscript. A comparison of all the samples is shown in **Fig. R26** below (**Supplementary Fig. 28**).

Fig. R26. A comparison of all the samples used in our manuscript.

We want to emphasize that while the sample used for the in-situ TEM nanomechanical testing contains some level of FIB damage, this does not undermine the reliability and accuracy of our conclusions. FIB damage is known to increase material brittleness¹⁶ or reduce SRO¹⁷. Nonetheless, our in-situ TEM nanomechanical testing aims to investigate how SRO is affected by mechanical deformation. We compare the SRO in the same sample region

before and after loading, thus the initial ductility or SRO level is not critically important.

We have included discussions regarding the implications of FIB damage on the results of **Fig. 5** in the supplementary materials (**Fig. R27-28**). **Figures R24 - R26** have also been added to the revised manuscript.

TEM sample preparation The samples underwent a TEM sample preparation procedure by lifting them out from the bulk sample and welding them on gold half-grids using the Thermo Fisher Scientific (TFS) Helios Nanofab 660 FIB. It is important to note that, for the LDED and LPBF samples, the lifting out of the sample was performed on the newest printed layer. This choice ensured that the selected region had not undergone any additional annealing processes. After the lift-out, the samples were thinned by Ga FIB working at 30 kV, which was subsequently reduced to 16 kV and then 8 kV. This process reduced the sample thickness to approximately 240 nm. The samples **used for Figs. 1-2 were** then flash polished^{56,57} to about 100 nm to remove the Ga⁺ damaged layer, *i.e.*, about 70 nm of thickness was removed from each side. Based on our Monte Carlo simulation of Ga⁺ ion implantation in CoCrNi, CoCrFeNi, and CoCrFeMnNi (**Supplementary Fig. 30**), the removed layer (~70 nm on each side) is much thicker than that of the damage layer induced by Ga⁺ ions during the FIB process (typically below 5 nm). The setup for flash polishing is shown in **Supplementary Fig. 31a**. The samples, held by gold tweezers, were immersed in 4% perchloric acid/ethanol solution at 12 V and around -45 °C. The polishing time was controlled by a timer and typically lasted between 50 ms and 200 ms, depending on the sample thickness. After flash polishing, the samples were washed three times in -45 °C methanol, -45 °C ethanol, and 25 °C ethanol, respectively. The resulting sample after flash polishing is shown in **Supplementary Fig. 31e-f**. **Note that the samples used for in-situ mechanical testing in Fig. 5 have not been flash-polished because it might compromise the stability of the mechanical testing process by etching the PTP structure or altering the crack shape, which is undesirable. The crack was deliberately created to localize the damage zone for a more precise capture of the SRO-deformation relationship. A discussion of the FIB damage in this sample used in Fig. 5 is provided in the Supplementary Notes F.**

Fig. R27. Update of the method section.

F. FIB damage on the sample used in Fig. 5

It is well-acknowledged that FIB samples inevitably contain dislocation loops on the surface due to Ga ion irradiation^{5,6}. In this work, we employed two methods for TEM sample preparation:

1. **FIB + Flash Polishing:** After FIB fabrication, we used a flash polishing method (detailed in our methods section) to remove the FIB-induced surface damage. The samples in **Figs. 1 and 2** were prepared using this approach.
2. **FIB Only:** This method was utilized for preparing the sample for in-situ TEM mechanical testing shown in **Fig. 5**. Flash polishing was not feasible here as it might compromise the stability of the mechanical testing process by etching the push-to-pull (PTP) structure or altering the crack shape, which is undesirable. The crack was deliberately created to localize the damage zone for a more precise capture of the SRO-deformation relationship.

Consequently, the difference in SRO relative intensity between **Figs. 1 and 5** stem from the distinct TEM sample fabrication methods. The FIB-only sample (**Fig. 5** in the manuscript) exhibits a weaker SRO signal than the flash-polished sample (**Fig. 2** in the manuscript), as the irradiation process can disrupt SRO⁷. This finding also highlights that irradiation can be an effective method to tune SRO, which is also mentioned in our discussion in the manuscript.

While the sample used for the in-situ TEM nanomechanical testing contains some level of FIB damage, this does not undermine the reliability and accuracy of our conclusions. FIB damage is known to increase material brittleness⁶ or reduce SRO⁷. Nonetheless, our in-situ TEM nanomechanical testing aims to investigate how SRO is affected by mechanical deformation. We compare the SRO in the same sample region before and after loading, thus the initial ductility or SRO level is not critically important.

Fig. R28. Update of the supplementary materials.

Finally, we would like to share an interesting digression. While FIB damage is not a significant factor in the study presented in **Fig. 5**, it plays a crucial role in numerous other experiments focused on the nanomechanical behavior of materials. In a separate project, we explored the removal of FIB damage through in-situ TEM annealing experiments conducted in a high vacuum, which we anticipated would preserve the sample's shape more effectively than flash polishing. However, this approach revealed an unexpected challenge. We found that chemisorbed oxygen on the sample's surface could easily trigger oxidation during the thermal annealing process. This resulted in the formation of complex oxide diffraction patterns, which in turn obscured the detection of SRO signals (refer to **Fig. R29** below). Although this observation will not be included in our manuscript, we thought you might find this digression interesting.

Redacted

Comment 4:

Also, CoNiV is not shown in Figure 2g, making it difficult to assess how relevant the small change in relative intensity shown in Figure 5e actually is (it seems to be less than some differences in Fig 2g).

Response:

We have included a new figure (see the modified **Fig. R26** below) in the Supplementary Materials to facilitate a comparison between **Fig. 2g** and **Fig. 5e**. As mentioned in our response to your comment #3, we employ two methods for fabricating TEM samples in our study. **Fig. R26** illustrates how both irradiation and mechanical deformation can effectively modulate SRO.

Fig. R26. A comparison of all the samples used in our manuscript.

Indeed, the variation in SRO caused by mechanical loading appears similar to that induced by thermal treatment. However, compared to thermal treatment and changing the solidification cooling rate, this is more controllable. With an increase in deformation loading cycles, the SRO reduces steadily. This observation is consistent for both the samples PTP#1 and PTP#2. We anticipate a further alteration in SRO when the loading cycle further increases.

We would like to emphasize that our in-situ TEM experiments are primarily proof-of-concept demonstrations. A thorough exploration of SRO modification through mechanical deformation is part of our future research and lies beyond the current paper's scope.

We also wish to mention the complexities of the in-situ TEM experiments presented in **Fig. 5**. These experiments are challenging due to the instability of the TEM holder and the risk of sample damage from slight shaking movements or bumps during holder insertion. We have conducted many repetitions of these experiments. And we also try to prevent the formation of planar defects for the better imaging of SRO. This necessitates using a suitable intermediate load – too small a load results in no observable change, while a larger load risk creating planar defects or fracturing the sample. The difficulty lies in the lack of precise control over sample dimensions using FIB, particularly regarding sample thickness, crack shape, and size, due to FIB's limited resolution and our aim to minimize irradiation damage. The variability in crack size and shape significantly influences the stress intensity factor and local strain rate, making no two FIB samples identical. Consequently, optimal loading parameters for one sample may not be suitable for another. We successfully replicated this experiment once more, but could not obtain data beyond 1000 cycles as the sample fractured. These experiments are also extremely time-consuming and costly, with each data point in **Fig. 5e** incurring thousands of dollars. We plan to delve deeper into this research area when more resources become available in the future.

We have added some discussion to improve our manuscript, as shown in **Fig. R30** below.

Mechanical tuning of SRO

To investigate additional methods that could be effectively integrated with thermal treatment and quench rate selection for enhanced SRO tuning, we have conducted in-situ nano-mechanical experiments using the EQ-SAED method. This allows us to elucidate the impact of mechanical deformation and dislocation slip on SRO. CoNiV was selected for this demonstration as it offers a better SNR in its SRO signal (**Fig. 2h**), which enables more precise quantification of the mechanical deformation effect. We fabricated a push-to-pull (PTP) setup from bulk CoNiV using a focused ion beam (FIB), as depicted in **Fig. 5 a,b**. This device offers a straightforward implementation of the tensile testing of a thin sample within the limited space of a TEM column, while ensuring stability throughout the loading process. **Figure 5c** illustrates that the EQ-SAED characterization was performed on the region circled ahead of the prefabricated crack tip at which the deformation concentrates. The loading curve is presented in **Fig. 5d**. It is important to note that the entire electron transparent region maintained a single-crystal structure without planar defects during the experiments. Previous studies have noted that SRO can be disrupted by dislocations gliding across it^{5,9,22}. In the case of cyclic loading, dislocations are repetitively emitted from the crack tip and migrate away from it during loading, and some of the dislocations are fully and partially recovered in the subsequent unloading. The net positive dislocation emission and gliding in each cycle allow us to study its effect on SRO. Aligning with this theoretical speculation, we observed a consistent decrease in relative SRO intensity as the number of cycles increased (**Fig. 5e**). A comparison of all the samples in this study is shown in **Supplementary Fig. 29**. Compared to thermal treatment and changing the cooling rate, mechanical tuning of SRO is more controllable.

Fig. R30. Update of the manuscript in Page 14.

Comment 5:

Finally, it would be helpful to provide better TEM images for the different samples, to demonstrate chemical and microstructural homogeneity and exclude any differences between the samples. From the low magnification images given in Figure 1 it is very difficult to assess what the different contrast features are. Also, it is quite surprising that the dislocation density is higher after annealing at 900°C. Therefore, a better discussion of the microstructure should precede the SRO analysis.

Response:

Thank you for highlighting the inadequacy of the magnification in **Fig. 1** and the necessity to establish the chemical and microstructural homogeneity of our samples more convincingly. The initial images were captured on the [112] zone axis, which may not offer the clearest contrast compared to two-beam condition images. They do, however, correspond with the diffraction patterns taken at the same axis. Recognizing that your concerns might echo those of other readers, we have diligently improved the manuscript as outlined below:

1. We have re-conducted TEM brightfield imaging for all samples in **Figs 1-2** at a higher magnification and under a two-beam condition near the [112] zone axis. This approach better highlights dislocation defects within the materials, as demonstrated in **Fig. R31**. The LPBF and LDED samples generally contain more dislocations than the other samples due to the fast cooling rates. Also, we observe dislocation cell wall structures that are typical of these additive-manufactured alloys¹⁸.

Fig. R31. Update of the Fig. 1 in the manuscript.

2. The microstructural homogeneity is evidenced in the TEM images (**Fig. R31**), XRD results (**Supplementary Fig. 2**), and SAED patterns (**Fig. R31**), all of which exhibit a single-phase FCC structure.
3. To confirm chemical homogeneity, we performed STEM-EDX elemental mapping for all samples at two different magnifications ($45,000\times$ and $350,000\times$), offering spatial resolution of 1.1 nm and 0.14 nm, respectively. All samples demonstrated chemical uniformity (**Figs. R32 - R40**), except for the CoCrFeMnNi fabricated via the LPBF method (**Fig. R41**), which exhibited some local segregation. We hypothesize that these non-equiatom local segregation areas might exhibit a varying degree of CSRO compared to the equiatom matrix. However, since our primary objective is to determine the presence (or absence) of CSRO at high cooling rates, rather than comparing the extent of order between equiatom and non-equiatom alloys, this variation in SRO does not affect the reliability of subsequent characterizations.

Fig. R32. STEM-EDX mapping for annealed CoCrNi.

Fig. R33. STEM-EDX mapping for as cast CoCrNi.

Fig. R34. STEM-EDX mapping for LDED CoCrNi.

Fig. R35. STEM-EDX mapping for LPBF CoCrNi.

Fig. R36. STEM-EDX mapping for as cast CoCrFeNi.

Fig. R37. STEM-EDX mapping for LDED CoCrFeNi.

Fig. R38. STEM-EDX mapping for LPBF CoCrFeNi.

Fig. R39. STEM-EDX mapping for as cast CoCrFeMnNi.

Fig. R40. STEM-EDX mapping for LDED CoCrFeMnNi. These small contaminations are CaO particles.

Fig. R41. STEM-EDX mapping for LPBF CoCrFeMnNi.

4. We selected the LPBF-produced CoCrNi for atomic-resolution STEM-HAADF and EDX imaging to demonstrate atomic-scale chemical and structural uniformity. This alloy was chosen due to its superior damage tolerance¹⁹ among the CoCrNi family and its rapid cooling rate, which might generate interest regarding its microstructure. As for the cast samples, their ability to achieve single-phase homogeneity is well-documented. Our findings, presented in **Fig. R42** below, confirm both structural and chemical uniformity.

Fig. R42. Atomic resolution STEM-HAADF and EDX mapping for LPBF CoCrNi, captured at [110] zone axis.

- We appreciate your observation regarding the annealed sample exhibiting more dislocations than the as-cast sample. These dislocations were induced by undue forces during polishing, initially intended to expedite oxide layer removal, causing a thicker plastically-deformed surface layer than other samples. We have improved our polishing approach for the annealed sample and prepared a new sample with new characterization. The results in **Fig. R31** show significantly fewer dislocations in the re-examined specimen. All the data, including the TEM image and the SRO characterization, has been updated based on the new annealed sample (**Figs. R43 – R44**).

Fig. R43. Screenshot showing the update of Fig. 1.

Fig. R44. Screenshot showing the update of Fig. 2.

6. Lastly, we would like to thank the reviewer #2 again for your helpful and professional comments which significantly help us improve our manuscript. The Fig. R45 below summarizes the changes we made in the manuscript and supplementary materials to address your comment #5.

To assess how cooling rates and thermal treatments influence SRO in MPEAs, we fabricate MPEAs with a cooling rate spanning seven orders of magnitude using traditional casting and several advanced techniques, including laser-directed energy deposition (LDED) and laser powder bed fusion (LPBF), as illustrated in Fig. 1a. The corresponding processing cooling rates^{7,39,40} are around 10^1 to 10^2 K s⁻¹, 10^3 to 10^5 K s⁻¹ and 10^5 to 10^7 K s⁻¹, respectively. Following processing, we also annealed the samples at 900 °C for seven days, a treatment recently reported to induce peak local ordering³³. Our objective is to contrast the SRO in these materials, thereby discerning the impacts of solidification quench rates, and thermal treatment, on SRO formation. Moreover, we have selected CoCrNi, CoCrFeNi, and CoCrFeMnNi for comparative analysis, potentially revealing insights into how compositional complexity influences SRO formation. These CoCrNi-based MPEAs were chosen based on their compelling mechanical performance, availability of large experimental datasets and versatile fabrication techniques^{41,42}. TEM, X-ray diffraction (XRD) and atomic-resolution STEM-EDX characterizations have verified that these CoCrNi-based MPEAs maintain a single phase and FCC structure even in the LPBF samples that were formed at the highest cooling rates (Fig. 1 and Supplementary Fig. 2 and Fig. 3). The LPBF and LDED samples generally contain more dislocations than the other samples due to the fast-cooling rates. Also, we observe dislocation cell wall structures that are typical of these additive-manufactured alloys⁴³. To confirm chemical homogeneity, we performed STEM-EDX elemental mapping for all samples at two different magnifications (45,000 X and 350,000 X), offering spatial resolution of 1.1 nm and 0.14 nm, respectively. All samples demonstrated chemical uniformity (Supplementary Fig. 4 to 12), except for the CoCrFeMnNi fabricated via the LPBF method (Supplementary Fig. 13), which exhibited some local clustering. We hypothesize that these non-equiatom local segregation areas might exhibit a varying degree of CSRO compared to the equiatom matrix. However, since our primary objective is to determine the presence (or absence) of CSRO at high cooling rates, rather than comparing the extent of order between equiatom and non-equiatom alloys, this variation in SRO does not affect the reliability of subsequent characterizations.

Fig. R45. Update of the manuscript on page 4.

Reviewer #3

Comment 1:

The authors examine the prevalence of short range order in CoCrNi and related higher-order MPEAs, using an “Enhanced Quantitative SAED” method. The alloys were prepared at 3 quench rates spanning ~ 7 orders of magnitude, and were also examined in the annealed condition. The authors convincingly demonstrate that SRO is present even in the 5 element MPEA prepared at the highest quench rate. Variations in compositional complexity and thermal preparation are shown to have very little impact on the relative degree of SRO, which is counter to most assumptions in the MPEA field. The authors go on to show that the degree of SRO can be changed via plastic deformation. This experimental work is quite well-presented and will have significant implications for the design of MPEAs.

Response:

We deeply appreciate your encouraging remarks regarding the presentation of our work and its potential impact on the design of MPEAs, mentioning that this work is “*quite well-presented and will have significant implications for the design of MPEAs.*”

Comment 2:

Based on their experimental observations, the authors describe a local diffusion, solidification, and remelting model for the development of SRO during solidification, which they demonstrate using MD simulations. Their simulations show that the SRO develops very quickly in both space and time; the transition zone from random solid solution to solid with SRO occurs within only 6 atomic layers, and SRO is observed at solidification rates up to 10^{11} K/s. I found this model to be much less convincing than the experimental work. My first inclination would be to hypothesize that some level of chemical ordering develops in the liquid prior to crystallization. Although I am primarily an experimentalist, I wonder if the lack of ordering in the simulated supercooled liquid and the rapid ordering in the simulated solid could be the result of the EAM potentials chosen for the simulations. The authors should address how the potentials were selected and whether they were optimized for the simulations of solidification.

Response:

Thank you for your thoughtful comments. We acknowledge your concerns regarding the possible influence of the selected EAM potentials on our simulation results, particularly in the context of ordering in the supercooled liquid and rapid ordering in the solid phase.

During solidification, we did observe the presence of local structure order within the supercooled liquid, as detailed in **Supplementary Fig. 22**. Measuring chemical short-range order in a liquid (or amorphous) state is inherently different (less straightforward) than in a crystalline structure. To address your query, we adopted the following methodology:

First, we compute the radial distribution function $g(r)$ in the liquid. The liquid regime is divided into 6 slabs according to their distance away from the solidification front (**Fig. R46a**). **Fig. R46b** shows the corresponding radial pair distribution functions in these six zones. The peaks in these regions shows location

independence. With the first peak distance, we then compute the number of i - j pairs (i and j type of atom). The corresponding results are shown in **Fig. R46c**. The number of chemical pairs in the six regimes remains the same, suggesting there is no strong chemical short-range ordering in the liquid region even close to the solidification front.

In response to your subsequent question regarding the choice of EAM potential, we conducted additional solidification simulations using a different EAM potential²⁰. This was done to validate our conclusion that “prevalent SRO can form during the solidification process, even at high cooling rates.” **Fig. R47** shows that no significant CSRO forms in the liquid prior to MPEA solidification.

We have added these new results and analysis in our manuscript and supplementary materials.

Fig. R46 (a) The liquid region (purple colored) in the solidification is divided into six slabs. Their radial pair distribution functions are shown in **(b)**. **(c)** The number of chemical pairs within the first nearest neighbor as a function of location in the liquid.

Fig. R47. Similar results using Choi et al. EAM potential. (a) The liquid region (purple colored) in the solidification is divided into six slabs. (b) Radial pair distribution functions of six slabs. (c) The number of chemical pairs within nearest neighbor as a function of location in the liquid.

CSRO formation during solidification is a compelling process that warrants further in-depth study. In the supercooled liquid (amorphous structure), we have found that, the same chemical SRO does not appear, while the structurally ordered local clusters, such as the icosahedron, exist (Supplementary Fig. 23-25, Supplementary Table 4-6 and Supplementary Notes D). Considering that crystal growth relies on the advancement of the solidification front (Fig. 3a), it is reasonable to hypothesize that CSRO predominantly emerges at phase transformation interface under non-equilibrium solidification conditions. We carefully examined the evolution of the solidification front by analyzing snapshots taken at 4 ps intervals (Fig. 4a,b). Specifically, Fig. 4a,b reveal the spatial and temporal evolution of structure and chemistry, respectively. The solidification front is rugged and exhibits heterogeneous movement along the growth direction. As time elapses, the segments on the interface move upward, indicating a liquid-solid phase transition. Intriguingly, some segments can retract by moving downward, indicating the existence of local remelting. This could occur as the newly solidified clusters, being thermodynamically unstable due to the presence of dangling bonds, reconfigure locally in search of more energetically favorable pairs, thus lowering the potential energy, as revealed by the black circles in Fig. 4b. To quantify the dynamic motion of the front, we measured the net movement distance per 4 ps along the interface (Fig. 4c). A positive value represents a net growth of solidification, while the opposite indicates remelting. Fig. 4c shows clearly that the solidification front moves in a heterogeneous manner. The segments of the solid-liquid interface that undergo crystal growth at the current moment may partially remelt in subsequent moments and *vice versa*. Yet, it is surprising that the detailed balance of local growth and remelting manages to constrain the roughness of the solidification front to just a few atomic layers, ensuring minimal variation in the interface height to reduce its surface area.

Fig. R48. Screenshot showing the update of the manuscript.

D. CSRO in the supercooled liquid

During solidification study, we did observe the presence of local structural order within the supercooled liquid, as detailed in **Supplementary Fig. 23**. Measuring chemical short-range order in a liquid (or amorphous) state is inherently different (less straightforward) than in a crystalline structure. To further clarify this, we adopted the following methodology:

First, we compute the radial distribution function $g(r)$ in the liquid. The liquid regime is divided into 6 slabs according to their distance away from the solidification front (**Supplementary Fig. 24a**). **Supplementary Fig. 24b** shows the corresponding radial pair distribution functions in these six zones. The peaks in these regions show location independence. With the first peak distance, we then compute the number of i - j pairs (i and j type of atom). The corresponding results are shown in **Supplementary Fig. 24c**. The number of chemical pairs in the six regimes remains the same, suggesting there is no strong chemical short-range ordering in the liquid region even close to the solidification front.

Regarding the choice of EAM potential, we conducted additional solidification simulations using a different EAM potential⁴. This was done to validate our conclusion that “prevalent SRO can form during the solidification process, even at high cooling rates.” **Supplementary Fig. 25** shows that no significant CSRO forms in the liquid prior to MPEA solidification.

Supplementary Fig. 24 | a, The liquid region (purple colored) in the solidification is divided into six slabs. Their radial pair distribution functions are shown in **b**, **c**. The number of chemical pairs within the first nearest neighbor as a function of location in the liquid.

Supplementary Fig. 25 | Similar results using Choi et al. EAM potential. a, The liquid region (purple colored) in the solidification is divided into six slabs. **b**, Radial pair distribution functions of six slabs. **c**, The number of chemical pairs within nearest neighbor as a function of location in the liquid.

Fig. R49. Screenshot showing the update of supplementary materials.

Comment 3:

Related to this model, on page 13 the authors indicate that the Ni-Ni bonds would be favored over Ni-Co or Ni-Cr, leading to local melting of the latter arrangements until the Ni in the vicinity is depleted. However, the enthalpy of mixing of Ni-Cr in particular is negative (as is Co-Cr, while that for Ni-Co is 0). In view of this, why would the Ni-Ni bonds be favored? Overall, the authors need to better justify how the small differences in the (negative) enthalpies of mixing could result in such rapid development of SRO during solidification.

Response:

We would like to thank the reviewer for the helpful comment. Our solidification model, using the model CoCrNi EAM potential, showed that Ni-Ni and Co-Cr bonds are favorable. To understand the thermodynamic origin of this ordering, we perform further and detailed analysis.

First, we compute the cohesive energy for all pairs (Ni-Ni, Co-Co, Cr-Cr, Ni-Co, Ni-Cr, Co-Cr), as shown in **Table R1** below. The corresponding order parameters are also shown in the right column. It can be seen that the most favored pair (Co-Cr) in the ternary alloy does not correspond to the lowest pair energy. We also compute the mixing enthalpy for the binary alloy, Co-Cr, Ni-Co, and Ni-Cr (**Table R2**). The Ni-Cr, showing a negative heat mixing, does not result in a favored pairing in the ternary alloy CoCrNi. We note that the classic way of computing mixing enthalpy (or cohesive energy) is using binary alloy, not considering the presence of the third element. Taking CoCrNi as an example, when computing the mixing enthalpy of Ni-Cr, the influence of Co has yet to be considered.

Table R1 Pair cohesive energy and pairwise order parameters

Pair	Cohesive energy (eV/atom)	Pairwise order parameter
Ni-Ni	-4.45 (lowest)	0.78 (second highest)
Co-Co	-4.41	-0.55
Cr-Cr	-3.89	-0.63
Ni-Co	-4.40	-0.50
Ni-Cr	-4.25	-0.25
Co-Cr	-4.24	1.04 (highest)

Table R2 The heat of mixing of binary random alloy

Binary random alloy	Heat of mixing (eV)
Ni-Co	0.0256
Ni-Cr	-0.0752
Co-Cr	-0.0952

To understand the driving force for Co-Cr and Ni-Ni ordering in ternary CoCrNi, we perform the insertion of Ni atom to Co-Cr alloys and see how the mixing enthalpy varies. **Fig. R50** shows the enthalpy changes when increasing the Ni concentration x in the $(\text{CoCr})_{1-x}\text{Ni}_x$ alloys. The addition of Ni to CoCr causes an increase in enthalpy, which indicates that CoCr is a favored pair. The ordering of CoCr will leave Ni clustering. **The results suggest that interpreting the chemical ordering in ternary alloys may need to consider the presence of all species for enthalpy calculation.**

Fig. R50. The effect of adding Ni to the CoCr on mixing enthalpy. The increase of Ni concentration x increases the enthalpy of $(\text{CoCr})_{1-x}\text{Ni}_x$ system.

To test this, we perform simulation and calculation using a different EAM potential²⁰. The spatial distribution of the order parameter for CoCrNi after annealing at 1,315 K is shown in **Fig. R51 a-b**. One can see that this potential gives rise to different results in which Ni-Cr, Co-Co, and Cr-Cr are favorable pairs. The binary mixing enthalpy (**Table R3**) indicates Ni-Co has the lowest negative value, and again, it could not interpret the ordering in ternary alloy. To understand the influence of the third element, we compute mixing enthalpy as a function of Co concentration x in $\text{Co}_x(\text{CrNi})_{1-x}$. The enthalpy increase with Co implies that Cr-Ni is favored, and Co-Cr and Co-Ni are unfavored, which aligns with the measured order parameters in the ternary alloy.

Table R3 The heat of mixing of binary random alloy using Choi EAM potential.

Binary random alloy	Heat of mixing (eV)
Ni-Co	-0.0176
Ni-Cr	0.0009
Co-Cr	0.0427

Fig. R51. (a, b) Atomic and pairwise order parameters in the annealed CoCrNi alloy using Choi et al. EAM potential. It is noted that Ni-Cr, Co-Co, and Cr-Cr are preferred pairings. (c) Adding Co increases the enthalpy of $\text{Co}_x(\text{CrNi})_{1-x}$ system.

Lastly, we used the second EAM potential and performed solidification modelling. Under the same fast cooling rate of 10^{11} K/s, a high degree of chemical short-range order has developed in the solidified region (**Fig. R52**). This indicates that the fast

atomic diffusion in the supercooled liquid matches or even surpasses the solidification rate and enables chemical ordering at a short timescale. This finding using the different EAM potential agrees with the main conclusion in our manuscript (i.e., “the diffusion of atoms in liquid is fast enough to endow the chemical ordering during rapid solidification”).

Fig. 52. (a, b) Atomic slices of the structure showing the distribution of atoms and pairwise order parameters after solidification, respectively.

We have added the results, analysis, and discussions in the revised manuscript and the supplementary materials (**Figs. R53 - R55**). The complex interplay between mixing enthalpy and ordering tendency raised by the reviewer is a fascinating aspect in multicomponent system that warrants further in-depth investigation.

We propose an atomistic mechanism for the CSRO formation during solidification, as schematically shown in **Fig. 4 g-j**. The randomly distributed atoms in the liquid adjacent to solid front diffuse and participate in the growth of solid. Due to the attractive interactions between atoms in the crystal solid, the favored pairs form and lead to chemical ordering, for instance, a precursor of Ni-Ni pair (**Fig. 4g**). When unfavorable atoms enter the first nearest neighbor shell of this Ni precursor (for instance, Co and Cr, as shown by the arrows in **Fig. 4h**), they can depart from the precursor either by interface diffusion or dissolution into liquid (*i.e.*, remelting), leaving room for extra Ni atoms to diffuse in and grow the preferable Ni-Ni pairs. The subsequent growth of Ni-Ni pairs depicts the formation of local chemical ordering of Ni-Ni (**Fig. 4i**). Once the Ni atoms within the accessible diffusion range are exhausted, the Co and Cr atoms eventually encapsulate the Ni-Ni nanocluster; thus, the chemical order cannot extend over a long range (**Fig. 4j**). The repeated local diffusion, solidification, and remelting (**Fig. 4 g-j**) act as the engine to sustain the local atomic reconfiguration at the solidification front, leading to the formation of CSRO even at a high cooling rate. While our atomistic simulations focus on the formation of SRO within the CoCrNi system, the findings bear broader implications when combined with our experimental results, extending their relevance to other MPEAs. The high concentrations of multiple principal elements provide the abundant sources required for chemical ordering, and the swift atomic diffusion realizes the process. Originating from the attractive/repulsive interactions among the constituent elements, the chemical order and its large extent are primarily determined by crystal growth velocity and the diffusivities of atoms. **A discussion about the thermodynamic origin of local ordering is provided in Supplementary Notes E and Supplementary Fig. 26-28.**

Fig. R53. Update of the manuscript in Page 12.

E. Thermodynamic origin of the order

Our solidification model, using the model CoCrNi EAM potential, showed that Ni-Ni and Co-Cr bonds are favorable. To understand the thermodynamic origin of this ordering, we perform further and detailed analysis.

First, we compute the cohesive energy for all pairs (Ni-Ni, Co-Co, Cr-Cr, Ni-Co, Ni-Cr, Co-Cr), as shown in **Supplementary Table 4**. The corresponding order parameters are also shown in the right column. It can be seen that the most favored pair (Co-Cr) in the ternary alloy does not correspond to the lowest pair energy. We also compute the mixing enthalpy for the binary alloy, Co-Cr, Ni-Co, and Ni-Cr (**Supplementary Table 5**). The Ni-Cr, showing a negative heat mixing, does not result in a favored pairing in the ternary alloy CoCrNi. We note that the classic way of computing mixing enthalpy (or cohesive energy) is using binary alloy, not considering the presence of the third element. For example, when computing the mixing enthalpy of Ni-Cr, the influence of Co has yet to be considered.

To understand the driving force for Co-Cr and Ni-Ni ordering in ternary CoCrNi, we perform the insertion of Ni atom to Co-Cr alloys and see how the mixing enthalpy varies. **Supplementary Fig. 26** shows the enthalpy changes when increasing the Ni concentration x in the $(\text{CoCr})_{1-x}\text{Ni}_x$ alloys. The addition of Ni to CoCr causes an increase in enthalpy, which indicates that CoCr is a favored pair. The ordering of CoCr will leave Ni clustering. **The results suggest that interpreting the chemical ordering in ternary alloys may need to consider the presence of all species for enthalpy calculation.**

To test this, we perform simulation and calculation using a different EAM potential⁴. The spatial distribution of the order parameter for CoCrNi after annealing at 1,315 K is shown in **Supplementary Fig. 27 a-b**. One can see that this potential gives rise to different results in which Ni-Cr, Co-Co, and Cr-Cr are favorable pairs. The binary mixing enthalpy indicates Ni-Co has the lowest negative value (**Supplementary Table 6**), and again, it could not interpret the ordering in ternary alloy. To understand the influence of the third element, we compute mixing enthalpy as a function of Co concentration x in $\text{Co}_x(\text{CrNi})_{1-x}$. The enthalpy increase with Co implies that Cr-Ni is favored, and Co-Cr and Co-Ni are unfavored, which aligns with the measured order parameters in the ternary alloy.

Lastly, we used second EAM potential and performed solidification modelling. Under the same fast cooling rate of 10^{11} K/s, a high degree of chemical short-range order has developed in the solidified region (**Supplementary Fig. 28**). This indicates that the fast atomic diffusion in the supercooled liquid matches or even surpasses the solidification rate and enables chemical ordering at a short timescale. This finding using the different EAM potential agrees with the main conclusion in our manuscript (i.e., “the diffusion of atoms in liquid is fast enough to endow the chemical ordering during rapid solidification”).

The complex interplay between mixing enthalpy and ordering tendency is a fascinating aspect in multicomponent system that warrants further in-depth investigation.

Fig. R54. Screenshot showing the update of the supplementary notes.

Supplementary Table 4. Pair cohesive energy and pairwise order parameters.

Pair	Cohesive energy (eV/atom)	Pairwise order parameter
Ni-Ni	-4.45 (lowest)	0.78 (second highest)
Co-Co	-4.41	-0.55
Cr-Cr	-3.89	-0.63
Ni-Co	-4.40	-0.50
Ni-Cr	-4.25	-0.25
Co-Cr	-4.24	1.04 (highest)

Supplementary Table 5. The heat of mixing of binary random alloy.

Binary random alloy	Heat of mixing (eV)
Ni-Co	0.0256
Ni-Cr	-0.0752
Co-Cr	-0.0952

Supplementary Table 6. The heat of mixing of binary random alloy using Choi EAM potential.

Binary random alloy	Heat of mixing (eV)
Ni-Co	-0.0176
Ni-Cr	0.0009
Co-Cr	0.0427

Fig. R54. Screenshot showing the update of the supplementary tables.

Reference

1. Ding, J., Yu, Q., Asta, M. & Ritchie, R. O. Tunable stacking fault energies by tailoring local chemical order in CrCoNi medium-entropy alloys. *Proc. Natl. Acad. Sci.* **115**, 8919-8924 (2018).
2. Zhang, R., Zhao, S., Ding, J., Chong, Y., Jia, T., Ophus, C., Asta, M., Ritchie, R. O. & Minor, A. M. Short-range order and its impact on the CrCoNi medium-entropy alloy. *Nature* **581**, 283-287 (2020).
3. Chen, X., Wang, Q., Cheng, Z., Zhu, M., Zhou, H., Jiang, P., Zhou, L., Xue, Q., Yuan, F., Zhu, J., Wu, X. L. & Ma, E. Direct observation of chemical short-range order in a medium-entropy alloy. *Nature* **592**, 712-716 (2021).
4. Walsh, F., Zhang, M., Ritchie, R. O., Minor, A. M. & Asta, M. Extra electron reflections in concentrated alloys do not necessitate short-range order. *Nat. Mater.* **22**, 926-929 (2023).
5. Coury, F. G., Miller, C., Field, R. & Kaufman, M. On the origin of diffuse intensities in FCC electron diffraction patterns. *Nature* **622**, 742-747 (2023).
6. Zhang, R., Zhao, S., Ophus, C., Deng, Y., Vachhani, S. J., Ozdol, B., Traylor, R., Bustillo, K. C., Morris, J. W., Chrzan, D. C., Asta, M. & Minor, A. M. Direct imaging of short-range order and its impact on deformation in Ti-6Al. *Sci. Adv.* **5**, eaax2799 (2019).
7. Wang, L., Ding, J., Chen, S., Jin, K., Zhang, Q., Cui, J., Wang, B., Chen, B., Li, T., Ren, Y., Zheng, S., Ming, K., Lu, W., Hou, J., Sha, G., Liang, J., Wang, L., Xue, Y. & Ma, E. Tailoring planar slip to achieve pure metal-like ductility in body-centred-cubic multi-principal element alloys. *Nat. Mater.* **22**, 950-957 (2023).
8. Li, Q. J., Sheng, H. & Ma, E. Strengthening in multi-principal element alloys with local-chemical-order roughened dislocation pathways. *Nat. Commun.* **10**, 3563 (2019).
9. Li, L., Chen, Z., Kuroiwa, S., Ito, M., Yuge, K., Kishida, K., Tanimoto, H., Yu, Y., Inui, H. & George, E. P. Evolution of short-range order and its effects on the plastic deformation behavior of single crystals of the equiatomic Cr-Co-Ni medium-entropy alloy. *Acta Mater.* **243**, 118537 (2023).
10. Rackwitz, J., Yu, Q., Yang, Y., Laplanche, G., George, E. P., Minor, A. M. & Ritchie, R. O. Effects of cryogenic temperature and grain size on fatigue-crack propagation in the medium-entropy CrCoNi alloy. *Acta Mater.* **200**, 351-365 (2020).
11. Potirniche, G., Horstemeyer, M., Gullett, P. & Jelinek, B. Atomistic modelling of fatigue crack growth and dislocation structuring in FCC crystals. *Proc. R. Soc. A: Math. Phys. Eng. Sci.* **462**, 3707-3731 (2006).
12. He, Q., Tang, P., Chen, H., Lan, S., Wang, J., Luan, J., Du, M., Liu, Y., Liu, C. & Pao, C. Understanding chemical short-range ordering/demixing coupled with lattice distortion in solid solution high entropy alloys. *Acta Mater.* **216**, 117140 (2021).
13. De Fontaine, D. The number of independent pair-correlation functions in multicomponent systems. *J. Appl. Crystallogr.* **4**, 15-19 (1971).
14. Owen, L. R., Playford, H. Y., Stone, H. J. & Tucker, M. G. A new approach to the analysis of short-range order in alloys using total scattering. *Acta Mater.* **115**, 155-166 (2016).
15. Kiener, D., Motz, C., Rester, M., Jenko, M. & Dehm, G. Fib damage of Cu and possible consequences for miniaturized mechanical tests. *Mater. Sci. Eng. A* **459**, 262-272 (2007).

16. Liu, J., Niu, R., Gu, J., Cabral, M., Song, M. & Liao, X. Effect of ion irradiation introduced by focused ion-beam milling on the mechanical behaviour of sub-micron-sized samples. *Sci. Rep.* **10**, 10324 (2020).
17. Arkoub, H. & Jin, M. Impact of chemical short-range order on radiation damage in Fe-Ni-Cr alloys. *Scr. Mater.* **229**, 115373 (2023).
18. Bertsch, K. M., Meric de Bellefon, G., Kuehl, B. & Thoma, D. J. Origin of dislocation structures in an additively manufactured austenitic stainless steel 316L. *Acta Mater.* **199**, 19-33 (2020).
19. Gludovatz, B., Hohenwarter, A., Thurston, K. V., Bei, H., Wu, Z., George, E. P. & Ritchie, R. O. Exceptional damage-tolerance of a medium-entropy alloy CrCoNi at cryogenic temperatures. *Nat. Commun.* **7**, 10602 (2016).
20. Choi, W.-M., Jo, Y. H., Sohn, S. S., Lee, S. & Lee, B.-J. Understanding the physical metallurgy of the cocrfemnni high-entropy alloy: An atomistic simulation study. *npj Comput. Mater.* **4**, 1 (2018).

REVIEWER COMMENTS

Reviewer #2 (Remarks to the Author):

see attached document

Reviewer #4 (Remarks to the Author):

As the editor has been unable to get in touch with the original referees (R#1 and #3), I have been asked to review, if the original concerns of reviewer #1 and reviewer #3 have been met and the manuscript is publishable at a technical level. I thus refrain from giving another full review. On any question raised by former Referees 1 and 3 I found well ordered and comprehensive answer. The manuscript has been revised appropriately adding additional discussions and supplementary material. I thus recommend the publication of this revised submission.

Reviewer #5 (Remarks to the Author):

A characterization and formation mechanism is presented for the presence of short-ranged order (SRO) in CoCrNi materials prepared under different solidification and thermal processing conditions. A measure of the SRO is provided through an augmented microscopy approach. In addition, further mechanical testing was performed to elucidate the presence of SRO compared to dislocation and grain boundary structures. The experimental measurements are supplemented by a coupled Monte-Carlo & Molecular Dynamics model. The model is used to illustrate the fossilization of SRO into the growing solid during rapid solidification. Lastly, it is discussed how different processing methods may allow mechanistic control of the SRO content.

The (modified) manuscript is well-written and provides reasonable evidence to support the claims/hypotheses made. The authors have well addressed the majority of concerns raised by previous referees, though some questions remain:

- In Fig. 1 is reported in the caption that the scale bar for bright-field TEM images is 500nm, but 100nm is indicated in the first panel of section b.
- Within the context of the microscopy method, when averaging over different Bragg peaks of a similar character, is there a bias that may be introduced arising from e.g. anisotropic strain? Would such contributions be negligible?
- To avoid further confusion about Fig. 2f, it may be helpful to add an indicator for which section of the experimental results the fitting had the most weight. Does this also explain why the fitting overestimates the higher radial distance?
- It is mentioned in section "Unraveling the mechanistic origin of SRO", that there is a maximum achievable SRO degree? How would one determine this, considering it has previously been argued that SRO is formed by kinetic frustration?
- To further expound on comment 2 of reviewer #3, does the SRO develop within the solid-liquid interface or deeper within the solidifying bulk?
- It would be beneficial to describe how the different EAM potentials compare/differ.
- An increase in the cooling rate has previously been correlated with an increase in defect density. Can the increase in SRO be well separated from the increase in other defects types and configurations?

An itemized list of responses to reviewers' remarks
(Blue: Reviewer's comments; Black: Our response)

Table of Contents

Reviewer #2 **2**

Reviewer #5 **11**

Reference **20**

(Blue: Reviewer's comments; Black: Our response)

Reviewer #2

Comment 1

The authors have given detailed comments to all the reviewers comments and further improved the manuscript.

Response:

We thank the reviewer for the encouraging comments on the improvements of our manuscript.

Comment 2

But they did not adequately resolve the main experimental issue, which is the fact that their SRO relative intensity measurement is erroneous. Instead they have tried to argue that they deliberately oversaturate the detector as this decreases the standard deviation (supplementary Fig. 17). This reasoning is wrong as a small standard deviation does not make a measurement approach correct.

In supplementary Fig. 16 they claim that they can recover the true height of the matrix peak by fitting just a part of the data outside an overexposed region. This is however not correct – fitting a gaussian peak to a part of the experimentally measured peak is not sufficient for obtaining the true peak height. The black line below is the example provided by the authors in supplementary Fig 16. – shifting the fit area (blue vs. red area) can change the measured intensity significantly.

In the same way fitting the peak after truncating it (the truncated peak is indicated by the solid black line and the true peak with the dashed peak) does not lead to an accurate result, even if the fit area is chosen next to the overexposed area. This can be seen below as the blue dotted fit deviates from the true peak.

Two side notes:

- (i) performing a gaussian fit with an offset makes the fit result more inaccurate as the offset determined by the fit changes the intensity that is detected
- (ii) a peak is not necessarily perfectly gaussian but can be also modeled using a Voigt profile making the fit even more ambiguous.

In conclusion, **the authors should simply repeat their measurement without overexposing the peaks.** If their detector does not provide sufficient dynamic range they can easily take multiple exposures.

Response:

We thank the reviewer for the helpful comments and especially appreciate the efforts in extracting data from our figure to perform some fitting. Not many reviewers are willing to undertake such a detailed analysis. We have carefully considered your comments and decided to repeat our experiments with an exposure time 10 times lower (0.1s) than in our previous experiments to avoid overexposure. We detail this update below. Please note that repeating all the experiments is not a trivial effort, as it is expensive and time-consuming. We believe that these additional efforts have significantly improved the reliability of our work.

A key summary of the new experiments is that we obtained the same conclusion as those in the previous version of our manuscript, with the details provided below.

Before presenting a detailed comparison of the new and old data, we wish to express our gratitude for the reviewer's persistence in repeating the experiments to obtain the un-overexposed data. This prompted us to re-examine our previous data, leading to the discovery that the *.TIFF file exported from the raw data using Velox software was corrupted. Consequently, many of the overexposure plateaus observed in our previous data were not due to overexposure; rather, they resulted from the data format conversion (see **Fig. R1** below). In our new analysis, we have directly used the *.MRC file, and we have resolved this issue. After this adjustment, the absolute value of the relative SRO intensity is corrected.

Fig. R1. The SAED image and the intensity line profile through the direct beam, displayed in various data formats and viewed with different software tools. The exposure time is 0.1 seconds. (a) Original data viewed with Velox software. (b) MRC file exported from Velox and viewed with ImageJ software. (c) TIFF file exported from Velox and viewed with ImageJ software, showing an artificial plateau in the line profile due to data format conversion errors.

Fig. R2 provides a comparative view of the previous and updated figures. The old dataset was captured with an **exposure time of 1 second**, whereas the new dataset utilized a shorter **exposure time of 0.1 seconds**. To improve the signal-to-

noise ratio (SNR), we collected 200 frames of SAED patterns per sample with the new exposure settings. By overlaying sets of 10 adjacent SAED patterns, we effectively enhanced the SNR, resulting in 20 composite images that simulate an aggregated exposure duration of 1 second, matching the effective exposure time used in the original experiment for a fair comparison.

Fig. R2. Comparative analysis between the old and new versions of Figure 2 in the manuscript. In the new data, the matrix peaks are correctly exposed, avoiding the overexposure observed previously. Additionally, peak heights are accurately determined using new fitting functions and the data format conversion issue has been resolved. The primary conclusions drawn from the initial data are reaffirmed by the results of the new experiments.

Fig. R3 provides a comparative view of the previous and updated **Supplementary Fig.27**.

Fig. R3. Comparative analysis between the old and new versions of Supplementary Fig. 27. The primary conclusions drawn from the initial data are reaffirmed by the results of the new experiments.

It is worth noting that we used the pseudo-Voigt fitting method for the matrix Bragg peaks. For the SRO, a Gaussian fitting is used. Both Gaussian and pseudo-Voigt fitting are commonly employed in peak fitting for diffraction patterns. The fitting ranges for SRO and Matrix peaks are $r \in [0, 30]$ pixels and $r \in [0, 25]$ pixels, respectively. As shown in **Fig. R2**, the fitting quality is excellent.

Regarding the in-situ TEM experiments presented in **Fig. 5** in the manuscript, it was not feasible to replicate the exact conditions using a reduced exposure time within several trials. As explained in our previous reply-to-reviewer version, these experiments are very challenging and time-consuming. For example, the samples can be easily broken before mechanical tests. Nevertheless, the use of the relative SRO intensity, which involves matrix peak intensity, may not be necessary for these in-situ TEM experiments. The consistency of the sampled area

for SAED imaging and the imaging conditions throughout the experiments lends confidence that a straightforward comparison of the SRO intensities is adequate.

We utilized the MRC files from the original data, which are unaffected by data conversion errors, to examine the evolution of the SRO peak intensity. Although some of the matrix peaks are overexposed, all the SRO peak's intensities are properly exposed as their intensities are 20-50 times lower than the matrix. We found that the SRO peak intensity decreases as the loading cycle increases, which agrees with the conclusion in our previous manuscript (see **Fig. R4** below).

Moreover, to substantiate our findings, we computed the ratio of SRO intensity to the overall background—the sum total of pixel intensities in a SAED pattern—and found that the outcomes are consistent with our initial results (see **Fig. R4** below).

Fig. R4. The evolution of the SRO intensity and the SRO/(overall background) of the sample used in Fig. 5. Data was obtained from the old data with 1s exposure time.

To facilitate the comparison with the data in **Figure 2** and **Figure 5** in the main text of the manuscript, such as the comparison shown in **Supplementary Fig. 26** and **Fig. R3**, we have applied a scaling factor α to the SRO intensity, thereby converting the SRO peak intensity to the relative SRO intensity. We first conducted a repeat SAED experiment on the same sample region using a 0.1s exposure time, where the SAED was properly exposed, to obtain the relative SRO intensity, denoted as $I_{0.1\text{ second}}^{1000\text{ cycles}}$. Then, the following formulae are used to calculate the scaling factor α :

$$I_{0.1\text{ second}}^{1000\text{ cycles}} = \frac{SRO_{0.1\text{ second}}^{1000\text{ cycles}}}{Matrix_{0.1\text{ second}}^{1000\text{ cycles}}} \quad (2)$$

$$\alpha = \frac{I_{0.1\text{ second}}^{1000\text{ cycles}}}{SRO_{1\text{ second}}^{1000\text{ cycles}}} \quad (3)$$

where the $Matrix_t^N$ and SRO_t^N are the Matrix and SRO peak intensity obtained from the data taken with an exposure time of t for the push-to pull (PTP) sample after N deformation cycles, respectively. Then the relative SRO intensity in Fig. 5 is computed using:

$$I_{1_{second}}^N = \alpha \times SRO_{1_{second}}^N, \quad N = 0, 500, 1000. \quad (4)$$

A comparison of the old and new **Figure 5** is shown below in **Fig. R5**. The same conclusion from the testing remains as “Aligning with this theoretical speculation, we observed a consistent decrease in relative SRO intensity as the number of deformation cycles increased”.

Fig. R5. Comparative analysis between the old and new versions of Fig.5. The primary conclusions drawn from the initial data are reaffirmed by the results of the new experiments and analysis.

Fig. R6-R9 below shows the additional updates on the manuscript and supplementary materials with changes highlighted in yellow.

To scrutinize the relative degree of SRO among the different samples more quantitatively, we have further improved the above SAED method, which is schematically illustrated in Fig. 2a-f. 200 SAED images were collected consecutively for each sample with the same imaging condition, such as a fixed exposure time and aperture size. Following this, a composite SAED pattern was constructed by overlaying 10 adjacent SAED patterns to increase the SNR, thereby achieving an aggregated exposure time of 1 second for later analysis. Then, the Bragg spots in each composite SAED image were detected by a cross-correlation algorithm, followed by the classification of Bragg spots into three categories: direct beam, matrix, and SRO. As shown in Fig. 2a, the accurate Bragg spot finding results are visualized by the red disks overlaid on the SAED patterns in Fig. 2a, sharing the same center as the Bragg peaks. Afterward, each SAED image is segmented into a series of square image tiles with the Bragg peak positioned at the center (Fig. 2 b,c). Furthermore, the tiles of the same type are averaged to enhance the SNR further, thus alleviating the need for an energy filter. For example, in each SAED pattern, the eight SRO-sensitive tiles closest to the direct beam are selected for averaging. Since there are 20 composite SAED images for each sample, thus a total of 160 tiles was used to obtain a single image of the SRO Bragg peak (Fig. 2d). The significantly improved SNR is clearly shown by comparing the raw images in Fig. 2c and processed images in Fig. 2d. To further quantitatively analyze the Bragg peak intensities, two-dimensional (2D) Gaussian fitting and pseudo-Voigt fitting are performed for SRO and Matrix peaks, respectively. The corresponding fitting results for both matrix peaks and SRO are shown in Fig. 2e, depicting excellent agreement with the experiments. To validate the quality of the 2D fitting analysis, a radial-integral with elliptical correction is performed on the 2D data (Fig. 2f), which again demonstrates the robustness of the method. Note that the selection of the beam current and exposure time in our experiments was a strategic decision aimed at enhancing the signal-to-noise ratio for the SRO signal and ensuring a close-parallel-beam condition. More detailed discussion can be found in Method and Supplementary Fig. 14. At last, the SRO Bragg peak intensity divided by the averaged matrix Bragg peak intensity, denoted as the “relative SRO intensity”, is computed to compare different MPEA samples with different compositions and thermal treatments (Fig. 2 g-h).

Fig. R6. Update of the manuscript on page 6.

TEM characterization and analysis The SAED patterns were collected in TFS Talos F2 200X2 at the same acquisition conditions on the same day, including an accelerating voltage of 200 kV, a spot size of 1, a screen current of 0.03 nA (22,500 X magnification, with the 40 select area aperture inserted), a select area aperture of 40 μm , a magnification of 22,500 \times , a camera length of 410 mm and an image resolution of 1024 \times 1024. The exposure times for the data in Fig. 2 and Fig. 5 are 0.1 s and 1 s, respectively. We have applied a scaling factor for obtaining the relative SRO intensity in Fig. 5, as explained in Supplementary Note 7 and Supplementary Fig. 29. The relationship between beam current and C2 strength for our TEM is measured and shown in Supplementary Fig. 14. The C2 strength was selected to approximate parallel beam conditions, ensuring sharper and more reliable SRO peaks. The details of the EQ-SAED method have been presented in the main text. The data processing was performed with the py4DSTEM package⁵⁹. The fitting ranges for SRO and Matrix peaks are $r \in [0, 30]$ pixels and $r \in [0, 25]$ pixels, respectively. The in-situ TEM mechanical experiments were performed with a Bruker pico-indenter (PI) 95 holder.

Fig. R7. Update of the methods section.

Fig. R8. Update of the Supplementary Figures 26 and 29.

Supplementary Note 7

The exposure times for the data in Fig. 2 and Fig. 5 are 0.1 s and 1 s, respectively. Some of the matrix peaks in the data for Fig. 5 have been over-exposed. Note that this in-situ TEM experiment was performed before the experiments shown in Fig. 2.

However, the use of relative SRO intensity, which involves matrix peak intensity may not be necessary for these specific in-situ TEM experiments. The consistency of the sampled area for SAED imaging and the imaging conditions throughout the experiments lends confidence that a straightforward comparison of the SRO intensities is adequate.

We utilized the MRC files from the original data to examine the evolution of the SRO peak intensity. Although some of the matrix peaks are overexposed, all the SRO peak's intensities are properly exposed as their intensities are 20-50 times lower than the matrix. We found that the SRO peak intensity decreases as the loading cycle increases.

Moreover, to substantiate our findings, we computed the ratio of SRO intensity to the overall background—the sum total of pixel intensities in a SAED pattern—and found that the outcomes are consistent with our initial results (see **Supplementary Fig. 29**).

To facilitate the comparison with the data in Fig. 2 and Fig. 5, we have applied a scaling factor α to the SRO intensity, thereby converting the SRO peak intensity to the relative SRO intensity. We first conducted a repeat SAED experiment on the same sample region using a 0.1s exposure time, where the SAED was properly exposed, to obtain the relative SRO intensity, denoted as $I_{0.1\text{ second}}^{1000\text{ cycles}}$. Then, the following formulae are used to calculate the scaling factor α :

$$I_{0.1\text{ second}}^{1000\text{ cycles}} = \frac{SRO_{0.1\text{ second}}^{1000\text{ cycles}}}{Matrix_{0.1\text{ second}}^{1000\text{ cycles}}} \quad (2)$$

$$\alpha = \frac{I_{0.1\text{ second}}^{1000\text{ cycles}}}{SRO_{1\text{ second}}^{1000\text{ cycles}}} \quad (3)$$

where the $Matrix_t^N$ and SRO_t^N are the Matrix and SRO peak intensity obtained from the data taken with an exposure time of t for the push-to pull (PTP) sample after N deformation cycles, respectively. Then the relative SRO intensity in Fig. 5 is computed using:

$$I_{1\text{ second}}^N = \alpha \times SRO_{1\text{ second}}^N, \quad N = 0, 500, 1000. \quad (4)$$

Fig. R9. Update of the supplementary notes 7.

Reviewer #5

Comment 1

A characterization and formation mechanism is presented for the presence of short-ranged order (SRO) in CoCrNi materials prepared under different solidification and thermal processing conditions. A measure of the SRO is provided through an augmented microscopy approach. In addition, further mechanical testing was performed to elucidate the presence of SRO compared to dislocation and grain boundary structures. The experimental measurements are supplemented by a coupled Monte-Carlo & Molecular Dynamics model. The model is used to illustrate the fossilization of SRO into the growing solid during rapid solidification. Lastly, it is discussed how different processing methods may allow mechanistic control of the SRO content.

The (modified) manuscript is well-written and provides reasonable evidence to support the claims/hypotheses made. The authors have well addressed the majority of concerns raised by previous referees, though some questions remain:

Response:

We thank the reviewer for the positive and encouraging comments, especially for pointing out that *“The (modified) manuscript is well-written and provides reasonable evidence to support the claims/hypotheses made. The authors have well addressed the majority of concerns raised by previous referees.”*

Comment 2

In Fig. 1 is reported in the caption that the scale bar for bright-field TEM images is 500nm, but 100nm is indicated in the first panel of section b.

Response:

Thanks for pointing out this issue. We have corrected it in the revised manuscript, as shown in **Fig. R10**.

Fig. 1 | Qualitative characterization of SRO in MPEAs with various thermal processing routes. a, Schematic of the as-cast, LDED, and LPBF sample preparation method. **b,** Bright-field TEM images of the selected area and SAED patterns under the same conditions for ten samples. The white arrows and circles indicate the diffused spots related to SRO. **c,** Schematic of the TTT diagram of SRO, showing the difference in SRO is limited even with a wide range of cooling rates. Scale bars for bright-field TEM images, 100 nm. Scale bars for SAED patterns, 5 nm⁻¹.

Fig. R10. Update of the figure caption on page 26.

Comment 3

Within the context of the microscopy method, when averaging over different Bragg peaks of a similar character, is there a bias that may be introduced arising from e.g. anisotropic strain? Would such contributions be negligible?

Response:

We acknowledge the possibility of anisotropic strain within our samples, which may contribute to either isotropic or anisotropic broadening of the Bragg peaks in selected area electron diffraction (SAED) patterns. While such strain could potentially bias the peak intensity measurements, we believe that these factors do not detract from the overarching conclusions of our research for the following two reasons:

1. The experiments were conducted multiple times across different samples, consistently yielding similar results. This suggests that the impact of anisotropic strain is minimal and does not significantly influence the observed trends.
2. The primary goal of this study is to conduct a qualitative assessment of the SRO levels, particularly to identify the presence or absence of SRO in samples formed at high cooling rates. As such, our focus is not on the quantitative measure but rather on the qualitative detection of SRO, which remains unaffected by the potential variations due to anisotropic strain. Therefore, the credibility of our characterizations related to the existence of CSRO under the conditions studied is maintained.

Comment 4

To avoid further confusion about Fig. 2f, it may be helpful to add an indicator for which section of the experimental results the fitting had the most weight. Does this also explain why the fitting overestimates the higher radial distance?

Response:

Thank you for pointing out the need to clarify the fitting range. The fitting ranges for SRO and Matrix peaks are $r \in [0, 30]$ pixels and $r \in [0, 25]$ pixels, respectively. We have added this in the method section, as shown in the screenshot below (**Fig. R11**).

TEM characterization and analysis The SAED patterns were collected in TFS Talos F2 200X2 at the same acquisition conditions on the same day, including an accelerating voltage of 200 kV, a spot size of 1, a screen current of 0.03 nA (22,500 X magnification, with the 40 select area aperture inserted), a select area aperture of 40 μm , a magnification of 22,500 \times , a camera length of 410 mm and an image resolution of 1024 \times 1024. The exposure times for the data in Fig. 2 and Fig. 5 are 0.1 s and 1 s, respectively. We have applied a scaling factor for obtaining the relative SRO intensity in Fig. 5, as explained in Supplementary Note 7 and Supplementary Fig. 29. The relationship between beam current and C2 strength for our TEM is measured and shown in Supplementary Fig. 14. The C2 strength was selected to approximate parallel beam conditions, ensuring sharper and more reliable SRO peaks. The details of the EQ-SAED method have been presented in the main text. The data processing was performed with the py4DSTEM package⁵⁹. The fitting ranges for SRO and Matrix peaks are $r \in [0, 30]$ pixels and $r \in [0, 25]$ pixels, respectively. The in-situ TEM mechanical experiments were performed with a Bruker pico-indenter (PI) 95 holder.

Fig. R11. Update of the method section.

The fitting overestimate problem is mainly caused by the data format conversion error and the overexposure (see more discussion in our reply to the 2nd comment of reviewer #2 in this file on **page 3**). In the revised manuscript, we have

solved these problems by re-performing all the experiments in Fig. 2 with an exposure time 10 times smaller than previous experiments. The updated Fig. 2 is also shown below (**Fig. R12**). The primary conclusions drawn from the old data are reaffirmed by the results of the new experiments.

Fig. R12. Comparative analysis between the old and new versions of Figure 2. In the new data, matrix peaks are correctly exposed, avoiding the overexposure observed previously. Additionally, peak heights are accurately determined using new fitting functions and the data format conversion issue has been resolved. The primary conclusions drawn from the initial data are reaffirmed by the results of the new experiments.

Comment 5

It is mentioned in section "Unraveling the mechanistic origin of SRO", that there is a maximum achievable SRO degree? How would one determine this, considering it has previously been argued that SRO is formed by kinetic frustration?

Response:

We would like to thank the reviewer for the comments on the degree of SRO. In the manuscript, we refer to the degree of SRO obtained from extended annealing as the ‘maximum.’

Fig. R13 illustrates the SRO parameter as a function of Monte Carlo (MC) swaps in the hybrid MC/MD aging simulations. It is noted that the SRO achieves saturation after a sufficient number of MC swaps, and we consider this saturation point as the ‘maximum’ value attainable through aging modeling.

Fig. R13. The variation of pairwise order parameters with MC swaps at 1,315 K.

Kinetic frustration often refers to the challenges that arise from high fluctuations in strain, defects, or composition that hinder a system's progression toward a thermodynamically favored state. This phenomenon has been thoroughly examined in a recent publication¹ by Professor Mitra Teheri from Johns Hopkins University, which delves into the specifics of kinetic frustration within CoCrNi medium entropy alloys.

We agree with the reviewer that kinetic frustration may impede increases in SRO in actual materials. Given that the MC/MD modeling can sample only a limited number of configurations, it is probable that the achieved SRO degree is not the absolute maximum, implying that the aged configuration may not represent the global minimum in the energy landscape. To clarify this point and avoid potential confusion, we now describe our annealed state as ‘equilibrium’ in the manuscript. This revision has been updated, as shown in **Fig. R14**.

To assess the proximity of the SRO after solidification to its equilibrium state of long-time annealing, we employed a hybrid Monte Carlo/molecular dynamic (MC/MD) approach to attain the equilibrium state. The results in Fig. 3e,f, show only a slightly higher degree of local CSRO is achieved after annealing, implying the chemical distribution has closely approached the equilibrium even at the fast solidification speed. In Fig. 3g, we compare the probability distributions of $\Delta\alpha_{\text{Ni-Ni}}$ and $\Delta\alpha_{\text{Co-Cr}}$ in the three systems, an ideal RSS, rapidly solidified state, and annealed equilibrium. With being zero for RSS, the mean values of $\Delta\alpha_{\text{Ni-Ni}}$ only increase from 0.71 (solidified sample) to 0.78 (aged) after aging. Similarly, the mean values of $\Delta\alpha_{\text{Co-Cr}}$ show a minor increase, from 0.95 in the solidified to 1.04 in the annealed sample (The probability distributions of the other four pairwise order parameters are shown in Supplementary Fig. 17). These results indicate that most of the CSROs are formed during the solidification process and that heat treatment cannot significantly improve the CSRO, even in the samples solidified at a cooling rate of 10^{10} K s^{-1} which was assumed to be RSS. It is important to recognize that kinetic frustration⁴⁶ could inhibit the enhancement of SRO in actual materials. Consequently, the level of SRO observed in our simulations may not represent the absolute maximum that can be achieved under different conditions or in a thermodynamically ideal system.

Fig. R14. Update of the manuscript on page 8.

Comment 6

To further expound on comment 2 of reviewer #3, does the SRO develop within the solid-liquid interface or deeper within the solidifying bulk?

Response:

The SRO develops in the solidification front, i.e., the interface between liquid and just solidified crystal. Fig. R15 below (i.e., Figure 4a-b in the manuscript) elucidates the formation process of SRO, occurring at the solid-liquid interface. The solidification front is rugged and exhibits heterogeneous movement along the growth direction. As time elapses, the segments on the interface move upward, indicating a liquid-solid phase transition (Fig. R15a). From the chemistry map (Fig. R15b), it can be seen that the chemical order (Ni-Ni and Co-Cr) is developed in the just solidified region. In the manuscript, we further discuss the atomic processes leading to SRO development, as shown in Figure 4g-j (Fig. R16).

Fig. R15. Snapshots of the solidification front evolution. The top and bottom rows show the structural and chemical characterizations, respectively.

Fig. R16. Figure 4 for your reference.

Comment 7

- It would be beneficial to describe how the different EAM potentials compare/differ.

Response:

We thank the reviewer for suggesting a comparison of the EAM potentials used in our paper.

The main figures in the manuscript used the EAM potential developed by Li et al², which is particularly calibrated to capture the energetics of stacking faults and chemical ordering in the Ni–Co–Cr system.

In response to comment 3 of reviewer #3, we have performed the same solidification simulations using a different EAM potential³. This potential was specifically developed to investigate the phenomena of sluggish diffusion and microtwinning at cryogenic temperatures within the CoCrFeMnNi high-entropy alloy system. Under the same fast cooling rate of 10^{11} K/s , a high degree of chemical short-range order has developed in the solidified region (**Fig. R17**). This indicates that the fast atomic diffusion in the supercooled liquid matches or even surpasses the solidification rate and enables chemical ordering at a short timescale. This finding using the different EAM potential agrees with the main conclusion in our manuscript (i.e., “the diffusion of atoms in liquid is fast enough to endow the chemical ordering during rapid solidification”).

Fig. R17. (a, b) The distribution of atoms and pairwise order parameters after solidification, respectively. The simulation is performed using a different EAM potential.

The following changes have been made in the supplementary materials to highlight further the difference between these two potentials (Fig. R18).

Supplementary Note 4

Our solidification model, using the model CoCrNi EAM potential, showed that Ni-Ni and Co-Cr bonds are favorable. This potential is particularly calibrated to capture the energetics of stacking faults and chemical ordering in the Ni-Co-Cr system. To understand the thermodynamic origin of this ordering, we perform further and detailed analysis.

First, we compute the cohesive energy for all pairs (Ni-Ni, Co-Co, Cr-Cr, Ni-Co, Ni-Cr, Co-Cr), as shown in **Supplementary Table 4**. The corresponding order parameters are also shown in the right column. It can be seen that the most favored pair (Co-Cr) in the ternary alloy does not correspond to the lowest pair energy. We also compute the mixing enthalpy for the binary alloy, Co-Cr, Ni-Co, and Ni-Cr (**Supplementary Table 5**). The Ni-Cr, showing a negative heat mixing, does not result in a favored pairing in the ternary alloy CoCrNi. We note that the classic way of computing mixing enthalpy (or cohesive energy) is using binary alloy, not considering the presence of the third element. Taking CoCrNi as an example, when computing the mixing enthalpy of Ni-Cr, the influence of Co has yet to be considered.

To understand the driving force for Co-Cr and Ni-Ni ordering in ternary CoCrNi, we perform the insertion of Ni atom to Co-Cr alloys and see how the mixing enthalpy varies. **Supplementary Fig. 23** shows the enthalpy changes when increasing the Ni concentration x in the $(\text{CoCr})_{1-x}\text{Ni}_x$ alloys. The addition of Ni to CoCr causes an increase in enthalpy, which indicates that CoCr is a favored pair. The ordering of CoCr will leave Ni clustering. **The results suggest that interpreting the chemical ordering in ternary alloys may need to consider the presence of all species for enthalpy calculation.**

To test this, we perform simulation and calculation using a different EAM potential⁴. This potential was specifically developed to investigate the phenomena of sluggish diffusion and microtwinning at cryogenic temperatures within the CoCrFeMnNi high-entropy alloy system. The spatial distribution of the order parameter for CoCrNi after annealing at 1,315 K is shown in **Supplementary Fig. 24 a-b**. One can see that this potential gives rise to different results in which Ni-Cr, Co-Co, and Cr-Cr are favorable pairs. The binary mixing enthalpy indicates Ni-Co has the lowest negative value (**Supplementary Table 6**), and again, it could not interpret the ordering in ternary alloy. To understand the influence of the third element, we compute mixing enthalpy as a function of Co concentration x in $\text{Co}_x(\text{CrNi})_{1-x}$. As shown in **Supplementary Fig. 24c**, the enthalpy increase with Co implies that Cr-Ni is favored and Co-Cr and Co-Ni are unfavored, which aligns with the measured order parameters in the ternary alloy.

Fig. R18. Update of the supplementary materials.

Comment 8

An increase in the cooling rate has previously been correlated with an increase in defect density. Can the increase in SRO be well separated from the increase in other defects types and configurations?

Response:

We thank the reviewer for the insightful comments.

Indeed, previous work has shown that an increased density of defects, such as stacking fault tetrahedrons and vacancies⁴, will develop in the samples with a high cooling rate.

The question raised—"Can the increase in SRO be clearly distinguished from the rise in other defect types and configurations?"—is profound and insightful. We would like to address it from two perspectives:

1. Physical Process Perspective:

It appears that the rise in SRO cannot be entirely isolated from the increase in other defect types and configurations. Recent findings suggest a correlation between SRO and stacking faults. For instance, Naghdi et al. observed a significant augmentation in chemical ordering within stacking fault regions using molecular dynamics simulations⁵. Similarly, experiments by Soel et al. have shown that mechanically induced SRO is directly associated with stacking faults⁶. This evidence supports the notion that the formation of SRO and certain defects may be interrelated phenomena.

2. Characterization Perspective:

Experimentally, the precise quantification of SRO remains a formidable challenge. A commentary in *Nature Materials* has indicated that stacking faults can also contribute to diffuse diffraction signals⁷. Moreover, due to the potential correlation between stacking faults and SRO, as noted earlier, a clear separation in characterization seems currently unattainable. Nevertheless, in our experiments, we have endeavored to select regions devoid of stacking faults for SAED analysis to facilitate the differentiation of SRO from defects to some extent. Given that our study's primary objective is to qualitatively assess the presence or absence of SRO in samples with high cooling rates, the precise separation of SRO from defects, while ideal, is not critical for our conclusions.

In simulations, we can precisely characterize SRO after rapid solidification and after annealing, as the atomic position of each element is known. The concordance between experimental results and simulations enhances the credibility of our conclusions.

The distinction between the increase in SRO and the concurrent rise in other defect types and configurations deserves further in-depth study.

We have added the above discussions in the manuscript and supplementary materials, as shown in **Fig. R19-20** below.

Additionally, understanding the impact of SRO on material properties necessitates a precise quantification of SRO due to the potentially non-monotonic nature of this relationship. A non-monotonic relationship suggests that small variations in SRO may significantly alter material properties in complex and potentially unpredictable ways. Therefore, without accurate measurement of SRO, it would be challenging to map this intricate relationship accurately, leading to misconceptions about the role of SRO. The EQ-SAED method we developed in this study is more quantitative than the previous EF-SAED method, and it negates the need for an energy filter, which considerably simplifies its application. However, this method remains semi-quantitative due to the inherent challenges in calibrating the exact degree of SRO. The precise quantification of SRO in MPEAs persists as a key hurdle in understanding the influence of SRO on material properties. It is crucial for future investigations to pioneer a more accurate quantification method, which could greatly expedite the exploration and application of diverse SRO modulation techniques. Such an advancement, though challenging, might be achievable through the integration of theoretical models grounded in a comprehensive understanding of interatomic interactions. In addition, the distinction between the increase in SRO and the concurrent rise in other defect types and configurations deserves further in-depth study. A discussion on this topic is provided in **Supplementary Note 5**.

In conclusion, the ubiquitous presence of SRO in MPEAs, attributable to its efficient formation during the solidification process across a wide cooling rate range, challenges the efficacy of thermal treatment as a method to modulate SRO. Consequently, future research should invest more in combination of multiple potential factors influencing SRO formation, as well as understanding the nuanced impacts of SRO on specific properties of MPEAs. This continued research is vital to unlocking new pathways to manipulate material properties and expanding our comprehension of these complex alloys.

Fig. R19. Update of the manuscript on page 14.

Supplementary Note 5

Previous work has shown that an increased density of defects, such as stacking fault tetrahedrons and vacancies⁵, will develop in the samples with a high cooling rate.

Can the increase in SRO be clearly distinguished from the rise in other defect types and configurations? We would like to address it from two perspectives:

1. Physical Process Perspective:

It appears that the rise in SRO cannot be entirely isolated from the increase in other defect types and configurations. Recent findings suggest a correlation between SRO and stacking faults. For instance, Naghdi et al. observed a significant augmentation in chemical ordering within stacking fault regions using molecular dynamics simulations⁶. Similarly, experiments by Soel et al. have shown that mechanically induced SRO is directly associated with stacking faults⁷. This evidence supports the notion that the formation of SRO and certain defects may be interrelated phenomena.

2. Characterization Perspective:

Experimentally, the precise quantification of SRO remains a formidable challenge. A commentary in *Nature Materials* has indicated that stacking faults can also contribute to diffuse diffraction signals⁸. Moreover, due to the potential correlation between stacking faults and SRO, as noted earlier, a clear separation in characterization seems currently unattainable. Nevertheless, in our experiments, we have endeavored to select regions devoid of stacking faults for SAED analysis to facilitate the differentiation of SRO from defects to some extent. Given that our study's primary objective is to qualitatively assess the presence or absence of SRO in samples with high cooling rates, the precise separation of SRO from defects, while ideal, is not critical for our conclusions.

In simulations, we can precisely characterize SRO after rapid solidification and after annealing, as the atomic position of each element is known. The concordance between experimental results and simulations enhances the credibility of our conclusions.

Fig. R20. Update of the supplementary materials.

Reference

1. Foley, D. L., Barnett, A. K., Rakita, Y., Perez, A., Das, P. P., Nicolopoulos, S., Spearot, D. E., Beyerlein, I. J., Falk, M. L. & Taheri, M. L. Diffuse electron scattering reveals kinetic frustration as origin of order in cocrni medium entropy alloy. *Acta Mater.* **268**, 119753 (2024).
2. Li, Q. J., Sheng, H. & Ma, E. Strengthening in multi-principal element alloys with local-chemical-order roughened dislocation pathways. *Nat. Commun.* **10**, 3563 (2019).
3. Choi, W.-M., Jo, Y. H., Sohn, S. S., Lee, S. & Lee, B.-J. Understanding the physical metallurgy of the CoCrFeMnNi high-entropy alloy: An atomistic simulation study. *npj Comput. Mater.* **4**, 1 (2018).
4. Takamura, J. Quenched-in vacancies and quenching strains in gold. *Acta Metall.* **9**, 547-557 (1961).
5. Naghdi, A. H., Karimi, K., Poisvert, A. E., Esfandiarpour, A., Alvarez, R., Sobkowicz, P., Alava, M. & Papanikolaou, S. Dislocation plasticity in equiatomic NiCoCr alloys: Effect of short-range order. *Phys. Rev. B* **107**, 094109 (2023).
6. Seol, J. B., Ko, W.-S., Sohn, S. S., Na, M. Y., Chang, H. J., Heo, Y.-U., Kim, J. G., Sung, H., Li, Z., Pereloma, E. & Kim, H. S. Mechanically derived short-range order and its impact on the multi-principal-element alloys. *Nat. Commun.* **13**, 6766 (2022).
7. Walsh, F., Zhang, M., Ritchie, R. O., Minor, A. M. & Asta, M. Extra electron reflections in concentrated alloys do not necessitate short-range order. *Nat. Mater.* **22**, 926-929 (2023).

REVIEWERS' COMMENTS

Reviewer #2 (Remarks to the Author):

The authors have repeated the experiments and as shown in Fig. R3. the corrected relative SRO intensities indeed differ very strongly from those presented in the original manuscript. Not only do the values differ by a factor more than 10, but additionally the relative order of different samples changed and the relative ratio of different samples is significantly different. For the in situ experiments the authors used a calibration factor to rescale for their erroneous measurements. The manuscript in its present form can be accepted, but to be more accurate the authors should not write that measuring the ratio is a quantitative measure of SRO, as e.g. stated in line 373.

Reviewer #5 (Remarks to the Author):

The authors have well addressed my comments and primary concern about how the fitting was performed. As such, I can recommend the article for publication after some minor comments:

- In reference to reviewer #5 comment 3, although the experiments were conducted across different samples, averaging may not remove a consistent strain broadening effect. Such behavior may arise from the presence of a given defect density found consistently between the samples.
- Is there a physical justification why a pseudo-Voigt fit is necessary for the "Matrix peaks" rather than Gaussian fit?

An itemized list of responses to reviewers' remarks
(Blue: Reviewer's comments; Black: Our response)

Table of Contents

Reviewer #2..... **2**

Reviewer #5..... **4**

(Blue: Reviewer's comments; Black: Our response)

Reviewer #2

Comment 1

The authors have repeated the experiments and as shown in Fig. R3, the corrected relative SRO intensities indeed differ very strongly from those presented in the original manuscript. Not only do the values differ by a factor more than 10, but additionally the relative order of different samples changed and the relative ratio of different samples is significantly different. For the *in situ* experiments the authors used a calibration factor to rescale for their erroneous measurements. The manuscript in its present form can be accepted, but to be more accurate the authors should not write that measuring the ratio is a quantitative measure of SRO, as e.g. stated in line 373.

Response:

We thank the reviewer for the suggestion to accept our manuscript and the efforts in the previous rounds of review which has helped us improve our manuscript.

Our EQ-SAED method is not fully quantitative, which has been previously mentioned in the *Discussion* section in our manuscript. However, our in-situ TEM experiments in Fig. 5 show that this method can to some extent reveal the predicted evolution of SRO at the crack tip during cycling loading. To make our language more precise, we have referred our method as a “semi-quantitative” method in the revised manuscript. The updated sentences are highlighted by yellow in our manuscript and shown in **Fig.R1**.

Abstract

Recent research in multi-principal element alloys (MPEAs) has increasingly focused on the exploration and exploitation of short-range order (SRO) to enhance material performance. However, the understanding of SRO formation and the precise tuning of it within MPEAs remains poorly understood, limiting the comprehension of its impact on material properties and impeding the advancement of SRO engineering. Here, leveraging advanced additive manufacturing techniques that produce samples with a wide range of cooling rates (up to 10^7 K s⁻¹) and an improved semi-quantitative electron microscopy method, we characterize SRO in three CoCrNi-based MPEAs to unravel the role of processing route and thermal history on SRO. Surprisingly, irrespective of the processing and thermal treatment applied, all samples exhibit

SAED methodology by integrating sophisticated data-processing techniques and a more robust experimental workflow. This method, named ESQ-SAED (Enhanced Semi-Quantitative SAED), not only obviates the need for a costly energy filter while maintaining a suitable signal-to-noise (SNR) ratio but also provides a promising avenue for more precise analysis. Our new method and the resultant characterization results are illustrated in the following.

manifesting a higher tendency to form chemical SRO¹⁴. Here, our observation of a higher degree of SRO in CoNiV is consistent with previous characterization results by other methods⁴⁵, indicating that our semi-quantitative characterization is reliable. Furthermore, our results suggest that choosing the constitutional element (*i.e.*, moderating the enthalpy term in Gibbs free energy) is more effective in tuning SRO than adjusting the number of constitutional elements.

Moreover, our experiment confirms the ability of the ESQ-SAED method to capture SRO information in MPEAs semi-quantitatively, further reinforcing its effectiveness and reliability. Recent studies have proposed that these diffuse signals at the $\frac{1}{2}$ $\{\bar{3}11\}$ location of FCC MPEAs electron diffraction on [112] zone axis (a key aspect of our ESQ-SAED approach) have alternative origins such as stacking faults⁴⁷ or HOLZ⁴⁸. In the case of stacking faults, we would expect an increase in diffuse signals during cyclic loading, due to the accumulation of planar defects introduced by plastic deformation. Conversely, if these signals were primarily a result of HOLZ, no significant changes should be observed during the cyclic loading of MEAs. Only the

Fig. 2 | ESQ-SAED characterization of SRO in MPEAs with various thermal processing routes. a-f, Illustration of the data processing procedure to enable more precise analysis of the degree of SRO. Each Bragg peaks in the SAED pattern (a) will be detected and labeled by the red dots in (b). The Bragg peaks will be further classified to create two sets of image tiles, one for the SRO and one for the matrix (c). The SNRs of the matrix and SRO peak are significantly improved by averaging the same type of image tiles, as shown in (d). Further, the fitting of the

Fig. R1. Update in the manuscript to address comment 1 from Reviewer #2.

Reviewer #5

Comment 1

In reference to reviewer #5 comment 3, although the experiments were conducted across different samples, averaging may not remove a consistent strain broadening effect. Such behavior may arise from the presence of a given defect density found consistently between the samples.

Response:

We thank the reviewer for pointing out this strain broadening effect. To verify if there is a consistent strain-broadening effect, we perform two further analyses of our data, as shown in **Supplementary Notes 8** and **Supplementary Fig.30-32**.

This schematic (**Fig. R2**) shows the definition of variables used in our analyses.

Fig. R2. Schematic showing the definition of variables used in our analyses.

First, we checked the full width at half maximum (FWHM) of the matrix peak for different types of samples (**Fig. R3a**). Although it was assumed that the samples with a higher cooling rate, such as the LDED and LPBF samples, may have a higher density of defects, we did not observe a more significant peak-broadening effect in these samples (**Fig. R3b**).

Fig. R3. **a.** Averaged FWHM of the matrix Bragg peak in different samples. **b.** Average of matrix FWHM for samples with different cooling rate.

Second, we used the integral of the peak to recalculate the relative SRO intensity. This method, using the integral of the peak (as shown in **Fig. R2**), calculates the shadow area under the peak curve, which can effectively ensure reliability even with the strain-led peak-broadening effect. The comparison between peak-value-based relative SRO intensity and integral-value-based relative SRO intensity is shown in **Fig. R4**. Our new analysis reveals the same conclusion as the one in our previous manuscript, showing that the strain-led peak-broadening has a neglectable effect on our conclusion.

Fig. R4. The comparison between peak-value-based relative SRO intensity and integral-value-based relative SRO intensity. a-b. Results of peak-value-based relative SRO intensity, i.e., the results shown in Fig. 2 in our paper. c-d. Integral-value-based relative SRO intensity.

py4DSTEM package⁵⁹. The fitting ranges for SRO and Matrix peaks are $r \in [0, 30]$ pixels and $r \in [0, 25]$ pixels, respectively. The anisotropic strain within our samples may contribute to either isotropic or anisotropic broadening of the Bragg peaks in SAED patterns; however, these factors do not detract from the overarching conclusions of our research, as discussed in **Supplementary Note 8 and Supplementary Fig. 30-32**. The in-situ TEM mechanical experiments were performed with a Bruker pico-indenter (PI) 95 holder.

Fig. R5. Update in manuscript on page 16.

Supplementary Note 8

Anisotropic strain within the samples may contribute to either isotropic or anisotropic broadening of the Bragg peaks in SAED patterns. While such strain could potentially bias the peak intensity measurements, we believe that these factors do not detract from the overarching conclusions of our research for the following two reasons:

1. The experiments were conducted multiple times across different samples, consistently yielding similar results. This suggests that the impact of anisotropic strain is minimal and does not significantly influence the observed trends.
2. The primary goal of this study is to conduct a qualitative assessment of the SRO levels, particularly to identify the presence or absence of SRO in samples formed at high cooling rates. As such, our focus is not on the quantitative measure but rather on the qualitative detection of SRO, which remains unaffected by the potential variations due to anisotropic strain. Therefore, the credibility of our characterizations related to the existence of CSRO under the conditions studied is maintained.

To verify if the strain-led peak-broadening effect is ignorable, we performed two further analyses of our data. The schematic (**Supplementary Fig. 30**) shows the definition of variables used in our analyses.

First, we checked the full width at half maximum (FWHM) of the matrix peak for different types of samples (**Supplementary Fig. 31a**). Although it was assumed that the samples with a higher cooling rate, such as the LDED and LPBF samples, may have a higher density of defects, we did not observe a more significant peak-broadening effect in these samples (**Supplementary Fig. 31b**).

Second, we used integral of the peak to recalculate the relative SRO intensity. This method, using the integral of the peak (as shown in **Supplementary Fig. 30**), calculates the shadow area under the peak curve, which can effectively ensure reliability even with the strain-led peak-broadening effect. The comparison between peak-value-based relative SRO intensity and integral-value-based relative SRO intensity is shown in **Supplementary Fig. 32**. Our new analysis reveals the same conclusion as the one in our previous manuscript, showing that the strain-led peak-broadening has a neglectable effect on our conclusion.

Fig. R6. Supplementary Notes 8 in SI.

Fig. R7. Added supplementary Fig. 30-32.

Comment 2

Is there a physical justification why a pseudo-Voigt fit is necessary for the “Matrix peaks” rather than Gaussian fit?

Response:

Thanks for your question.

- 1) The use of fit is actually not necessary for matrix peaks, but we performed the fit with the aim of making the analysis method for Matrix and SRO consistent. The reasons why the use of fit is not necessary for matrix peaks are provided below:

In our manuscript, the peak fitting is used to more accurately determine the peak intensity. We used the Gaussian function and the pseudo-Voigt function to fit the SRO and Matrix peaks, respectively.

For SRO peaks that are very weak and suffer from a low signal-to-noise ratio (SNR) issues, the use of fitting can effectively enhance the accuracy and reliability of the identification of peak intensity.

However, for the matrix peak, which is bright and has a high SNR, there is actually no need to do the fitting. For the matrix, the peak intensity found from the fitted curve is almost the same as the maximum value of the peak in the raw data.

- 2) We hope to further elaborate on the reason for choosing different functions for the fitting.

Pseudo-Voigt function rather than a Gaussian is more often selected for fitting the Bragg peaks, as the peaks typically exhibit a mixture of Gaussian and Lorentzian characteristics due to various scattering processes:

- When diffraction occurs from a crystal, Bragg peaks are convolved with a "shape factor". The resultant peak profiles can vary significantly depending on the particle size.
- The diffraction peaks incorporate both elastic and inelastic scattering components. The inelastic losses, particularly from bulk and surface plasmons, are predominantly Lorentzian, causing broader, longer-tailed shapes.

The pseudo-Voigt function, which is a linear combination of Gaussian and Lorentzian functions, offers a flexible model that can accurately represent the mixed character of the peaks. It allows for weighting between Gaussian and Lorentzian shapes, making it particularly suited for fitting Bragg peaks where these mixed characteristics are evident.

Specifically, for Matrix peaks which derive predominantly from the face-centered cubic (fcc) matrix lattice structure, the significant inelastic losses lead to a more Lorentzian-like shape, justifying the pseudo-Voigt approach over a pure Gaussian. The inelastic scattering caused by SRO is much weaker due to its low volume fraction. Thus, Gaussian fitting is already good for the SRO peaks.